



# Vertical structure of a springtime smoky and humid troposphere over the Southeast Atlantic from aircraft and reanalysis

Kristina Pistone[1,2], Eric M. Wilcox[3], Paquita Zuidema[4], Marco Giordano[3], James Podolske[2], Samuel E. LeBlanc[1,2], Meloë Kacenelenbogen[5], Steven G. Howell[6], and Steffen Freitag[6,7]

[1]Bay Area Environmental Research Institute, Moffett Field, CA, USA

[2]NASA Ames Research Center, Moffett Field, CA, USA

[3]Desert Research Institute, Reno, NV, USA

[4]Department of Atmospheric Sciences, Rosenstiel School of Marine, Atmospheric and Earth Science, University of Miami, Miami, FL, USA

[5]NASA Goddard Space Flight Center, Greenbelt, MD, USA

[6]University of Hawaii at Manoa, Honolulu, HI, USA

[7]now at: State Agency for Nature, Environment and Consumer Protection in North Rhine- Westphalia (LANUV NRW), Recklinghausen, Germany

**Correspondence:** Kristina Pistone (kristina.pistone@nasa.gov)

**Abstract.** The springtime atmosphere over the southeast Atlantic Ocean (SEA) is subjected to a consistent layer of biomass burning (BB) smoke from widespread fires on the African continent. An elevated humidity signal is co-incident with this layer, consistently proportional to the amount of smoke present. The combined humidity and BB aerosol has potentially significant radiative and dynamic impacts. Here we use aircraft-based observations from the NASA ORACLES (ObseRvations of Aerosols

above CLouds and their intEractionS) deployments in conjunction with reanalyses to characterize co-variations in humidity and BB smoke.

The observed plume-vapor relationship, and its agreement with the ERA5 and CAMS reanalyses, persists across all observations, although the magnitude of the relationship varies as the season progresses. Water vapor is well-represented by the reanalyses, while CAMS tends to underestimate carbon monoxide especially under high BB. While CAMS aerosol optical

depth (AOD) is generally overestimated relative to ORACLES AOD, the observations show a consistent relationship between CO and aerosol extinction, demonstrating the utility of the CO tracer to understanding vertical aerosol distribution.

We next use k-means clustering of the reanalyses to examine multi-year seasonal patterns and distributions. We identify canonical profile types of humidity and of CO, allowing us to characterize changes in vapor and BB atmospheric structures, and their impacts, as they covary. Predominant profile types vary spatiotemporally across the SEA region and through the

season. With this work, we establish a framework for a more complete analysis of the broader radiative and dynamical effects of humid aerosols over the SEA.



# 1  Introduction

Aerosol effects on atmospheric radiative transfer and dynamics are complex and varied. Aerosols not only produce direct radiative heating or cooling, but may also alter the properties of nearby clouds by directly changing a cloud's reflectivity (albedo), thickness, or altitude; the total cloud amount; or the surrounding atmospheric dynamics. These effects are especially complex for absorbing aerosols such as biomass burning (BB) smoke. The combination of direct, indirect, and semidirect effects together may result in either an overall warming or cooling effect (e.g., Koch and Del Genio, 2010) depending on the specific aerosol properties, but also on the surrounding cloud, radiative, and meteorological context. A key consideration to understanding this aerosol-cloud-climate puzzle is the role of atmospheric water vapor, both in total amount and its location. Water vapor plays a significant role as a climate feedback parameter (e.g., Forster et al., 2021), has its own localized radiative heating effects, and is important to cloud formation and lifetime in the present and future climate. Thus, it is important to consider how the local atmospheric properties may either affect or be affected by the presence of aerosols and water vapor together.

The atmosphere over the southeast Atlantic Ocean (SEA) is subjected to a consistent layer of biomass burning (BB) smoke from widespread fires on the African continent in the austral springtime (August through October). This smoke layer is initially lofted high in a continental mixed layer ($\sim$5-6km, $\sim 500 - 600$hPa) before being transported westward by the south African Easterly Jet (AEJ-S, $\sim$600-700hPa) where it overlies and ultimately mixes into the SEA stratocumulus-topped oceanic boundary layer. These conditions make it an ideal location to study the varied aerosol effects on climate. The NASA ORACLES (ObseRvations of Aerosols above CLouds and their intEractionS; Redemann et al., 2021) campaign was designed to study just that; in the present work, we use aircraft-based observations from the three ORACLES deployments, which, by design spanned the spring BB season (September 2016, August 2017, and October 2018). Taken together, these data allow us to characterize the spatial and temporal variations in atmospheric chemistry, aerosol, and meteorological structure as the season progresses, specifically the co-variations in humidity and aerosol. In a previous work (Pistone et al., 2021), we showed how observations from ORACLES-2016 exhibited a strong correlation between plume strength (i.e., CO concentration and aerosol extinction) and water vapor specific humidity. This is consistent with previous work over the region; for example, (Adebiyi et al., 2015) had analyzed MODIS and radiosonde data (over St Helena Island, 15.9°S, 5.6°W) and found that increases in aerosol optical depth were associated with increases in mid-troposphere (700-500hPa) moisture content, and the presence of humidity was noted even in southern African BB aerosol measurements in the Southern African Regional Science Initiative 2000 (SAFARI 2000) campaign (e.g. Haywood et al., 2003; Magi and Hobbs, 2003). More recently, Cochrane et al. (2022) used high-vertical-resolution aircraft data to quantify the aerosol and water vapor heating rates for specific case studies from ORACLES-2016 and -2017. Both these studies concluded that there is significant shortwave water vapor heating coincident with the aerosol heating within the combined aerosol/humidity layer, despite longwave water vapor cooling.

The presence of humid smoke may have additional impacts on the lower atmosphere, including on the underlying stratocumulus clouds. Absorbing aerosols above the cloud layer have been found to suppress cloud-top turbulent fluxes, resulting in physically thicker clouds with higher liquid water path (LWP) (e.g., Wilcox, 2010; Wilcox et al., 2016) and thus higher cloud




albedo. In a modeling-based study, Ackerman et al. (2004) modeled the influence of above-cloud water vapor using several case studies informed by field measurements, and concluded that the cloud liquid water path response to aerosol (i.e., increasing LWP due to precipitation suppression) had a much stronger response in the presence of overlying water vapor than under dry conditions, due to changes in cloud-top entrainment. Deaconu et al. (2019) used satellites and reanalysis to conclude that

absorbing aerosols in the SEA increased cloud optical thickness and liquid water path, and lowered cloud top altitudes.

Pistone et al. (2021) also presented evidence that the smoke/humidity relationship is present at the air mass origin over the African continent, thus indicating that these airmasses are likely subjected to the combined effects of humidity and BB aerosols for an extended time. In other words, the humid atmosphere observed over the SEA is subjected to not just instantaneous but cumulative direct and semi-direct effects of the layer of BB aerosol, from the time it took a given airmass to leave the

continent to when it was measured by the ORACLES flights (on the order of a few days to weeks from origin to observation), above and into the cloud layer. Within our current context, it is important to recognize that the humidity of the above-cloud air will influence the magnitude of aerosol-forced dynamics effects on clouds. From a practical measurement standpoint, humid aerosols also undergo swelling which affects their remote sensing retrievals and comparisons to modeled results (e.g., Shinozuka et al., 2020; Doherty et al., 2022). By combining aircraft data and large-scale reanalyses we aim to characterize

the variations in water vapor in conjunction with BB aerosol over the springtime southeast Atlantic region which allows us to assess the realism of the reanalysis datasets commonly used to initialize and constrain model simulations.

There are three major goals of this paper. First, to expand the results of Pistone et al. (2021) to discuss how the observed patterns in water vapor and CO vary across the three ORACLES observation periods, which by design sampled different times during the BB season. Second, to assess how well selected reanalysis products capture the results in the observations. Third,

to describe a new method to use climatological data to more completely characterize the atmospheric structure over this region and its radiative impacts.

Section 2 describes the observations and reanalysis used in this work. In Section 3 we extend the findings of Pistone et al. (2021) to describe the observations from the three deployment years, including a new discussion of CAMS and the climatological representativity of its BB tracers, specific humidity, and aerosol parameters. In Section 4, we then describe our use of a

k-means clustering of the ERA5 and CAMS reanalyses to identify canonical atmospheric profile types of varying atmospheric specific humidity ($q$), carbon monoxide (CO), and vertical structure and describe their changing (co)incidence spatially and throughout the season. Section 5 discusses the broader significance of this work within the context of previous works on this topic. With this work, we more completely describe the atmospheric structure of smoke and humidity over the SEA and lay out a framework for further analysis of the radiative heating of this humid plume within this region, and its effects on clouds

and climate over time and space.

## 2   Aircraft data and reanalyses

In the present work, we use aircraft observations from ORACLES, in conjunction with fields from the ERA5, CAMS, and MERRA-2 reanalyses.



## 2.1 Aircraft instrumentation

The ORACLES field mission consisted of three deployments: September 2016 out of Walvis Bay, Namibia; August 2017 out of São Tomé, São Tomé e Príncipe; and October 2018 also out of São Tomé. All instruments used here were deployed on the NASA P-3 aircraft during all three ORACLES deployments. 1 Hz measurements are used unless otherwise indicated. A

detailed description of the ORACLES experimental construction and instrumentation used here may be found in Redemann et al. (2021), or in the archived instrument dataset metadata (See Data Availability).

Unless otherwise noted, the present analysis focuses on aircraft profile data during each ORACLES deployment, with the 1Hz inlet-based aircraft data averaged within 100m ($\pm$50m) of the reanalysis pressure levels. The locations of these profile data are shown in Figure 1a-c, showing the locations of 105, 70, and 80 full or partial profiles (ramps or square spirals) sampled

during the 2016, 2017, and 2018 campaigns, respectively. Spatial subdivisions used in previous works, as well as the current framework, are also illustrated in Figure 1d. Due to the more northern deployment of ORACLES-2017 and -2018, the data shown in Section 3 are subselected to observations south of $5°$S, to isolate airmasses influenced by the AEJ-S (Pistone et al., 2021).

### 2.1.1 Inlet-based carbon monoxide and water vapor

In all ORACLES deployments, volume mixing ratios of carbon monoxide (CO), carbon dioxide ($CO_2$), and water vapor ($q$) were measured by a Los Gatos Research $CO/CO_2/H_2O$ Analyzer (known as COMA), modified for flight operations. It uses off-axis integrated cavity output spectroscopy (ICOS) technology to make stable cavity enhanced absorption measurements of CO, $CO_2$, and $H_2O$ in the infrared spectral region, technology that previously flew on other airborne research platforms with a precision of 0.5 ppbv over 10s (Liu et al., 2017; Provencal et al., 2005). Water vapor measurements of less than 50 ppmv

($\sim$0.03 g/kg) were removed due to instrument limitations, but this has minimal effect on the data considered here.

In Pistone et al. (2021), we discussed the validation of COMA measurements against two other onboard water vapor and inlet-based gas/aerosol instruments on the payload; following the good agreement shown in that work, the specific humidity measurements presented here are from COMA. We focus on carbon monoxide as a BB tracer because it is a more straight-forward parameter to accurately model, and, as will be shown, agrees well with observed aerosol extinction across the three

deployments.

### 2.1.2 Aerosol optical depth

The Spectrometer for Sky-Scanning Sun-Tracking Atmospheric Research (4STAR; Dunagan et al., 2013) is an airborne hyperspectral (350-1700 nm) sun photometer which measures direct-beam solar irradiance (sun-tracking mode) for retrieval of column aerosol optical depth (AOD; e.g., Shinozuka et al., 2013) and column trace gases (e.g., Segal-Rosenheimer et al., 2014)

above the aircraft level. This work presents the column AOD (LeBlanc et al., 2020); we note that Pistone et al. (2021) also utilized 4STAR's column water vapor (CWV) measurements in a similar analysis.





### 2.1.3 Aerosol extinction

The Hawaii Group for Environmental Aerosol Research (HiGEAR) operated several in-situ aerosol instruments on the P-3. Total and sub-micrometer aerosol light scattering coefficients ($\sigma_{\mathrm{scat}}$) were measured onboard the aircraft using two TSI model 3563 3-wavelength nephelometers (at 450, 550, and 700 nm) corrected according to Anderson and Ogren (1998). The scattering used here is from TSI Nephelometer #1 (as identified in the ORACLES dataset) interpolated to the PSAP absorption wavelengths (below). The #1 Nephelometer measured total aerosol scattering Aerosol extinction is the sum of PSAP absorption plus nephelometer scattering.

Light absorption coefficients ($\sigma_{\mathrm{abs}}$) at 470, 530, and 660 nm were measured using two Radiance Research particle soot absorption photometers (PSAPs). We use absorption from the "Front" PSAP. The humidity within the PSAP was not explicitly controlled, but the PSAP optical block was heated to approximately 50°C to reduce artifacts which would result from a changing RH; this had the effect of reducing relative humidity in this instrument to much lower than the 40% within the nephelometers. The PSAP absorption was corrected using the wavelength-averaged correction algorithm (Virkkula, 2010) following the conclusions of Pistone et al. (2019). Instrument noise levels are 0.5 Mm$^{-1}$ for a 240–300 s sample average, comparable to values reported previously (Anderson et al., 2003; McNaughton et al., 2011).

### 2.2 Large-scale reanalyses

In considering the reanalyses, we use specific humidity ($q$), carbon monoxide (CO), and aerosol optical depth (AOD) fields for the following subsets and resolutions.

The European Centre for Medium-Range Weather Forecasts (ECMWF) has developed global atmospheric reanalysis products for several decades, with ERA5 (ECMWF Reanalysis version 5) being the current product (Hersbach et al., 2019). We consider ERA5 at 0.25-degree, hourly resolution, at pressure levels of 50hPa intervals between 400 to 750hPa and 25hPa intervals from 750 to 1000hPa in the comparison with ORACLES flights, and 0.25-degree, 3-hourly resolution in the k-means analysis. ERA5 does not report atmospheric chemistry or aerosols nor does it directly incorporate aerosol effects, though satellite measurements of aerosol-influenced radiances are incorporated into the reanalysis. In a slight difference from Pistone et al. (2021), here we add the 400hPa level (approx 7.6km) to our analysis to better align with the CAMS reanalysis (below) which reports at 400hPa rather than 450hPa; our previous work only extended to 450hPa (roughly 6.7km), which was chosen to capture the full BB plume height and ORACLES P-3 aircraft operating altitudes.

For analysis of aerosol/chemistry parameters, we consider the Copernicus Atmosphere Monitoring Service (CAMS) reanalysis. CAMS includes chemistry but is coarser than ERA5 both spatially and temporally, namely 3-hourly versus hourly; 0.75° versus 0.25°, and 10 pressure levels from the surface to 400hPa rather than 18 levels (100hPa intervals in the plume level). CAMS uses the same meteorological inputs as does ERA5, assimilates MOPITT (on NASA's Terra satellite) CO retrievals (among other chemistry), and includes radiatively-active aerosol fields (Inness et al., 2019). While total AOD (from MODIS on both NASA's Terra and Aqua satellites, for our study period) is assimilated into CAMS from satellite observations, the





model aerosol scheme uses twelve speciated aerosol tracers, which likely leads to less-constrained aerosol results compared to those for trace gases (Inness et al., 2019).

The Modern-Era Retrospective Analysis for Research and Applications, version 2 (MERRA-2) is an atmospheric reanalysis produced by NASA's Global Modeling and Assimilation Office (GMAO) (Gelaro et al., 2017; Randles et al., 2017; Buchard et al., 2017). MERRA-2 assimilates observations of meteorological parameters from multiple satellite platforms, as well as aerosol optical depth from satellites (MODIS, AVHRR) and ground-based (AERONET) measurements, into a comprehensive atmospheric model, with explicit accounting of aerosol radiative effects. MERRA-2 datasets are given on a nominal 50 km horizontal resolution ($0.5° \times 0.5°$) with 72 vertical layers from the surface to 0.01 hPa. Our analysis here subsets this product to the same ERA5 pressure levels for ease of interpretation. The complete set of MERRA-2 files have been sampled up to 1-second resolution along every ORACLES flight (Collow et al., 2020); we use this product averaged to CAMS and ERA5 altitudes to facilitate the comparison. While the focus of latter sections is on CAMS and ERA5, MERRA-2 results from 2016 only were discussed in Pistone et al. (2021); we thus discuss this product in Section 3 for completeness over all ORACLES deployments.



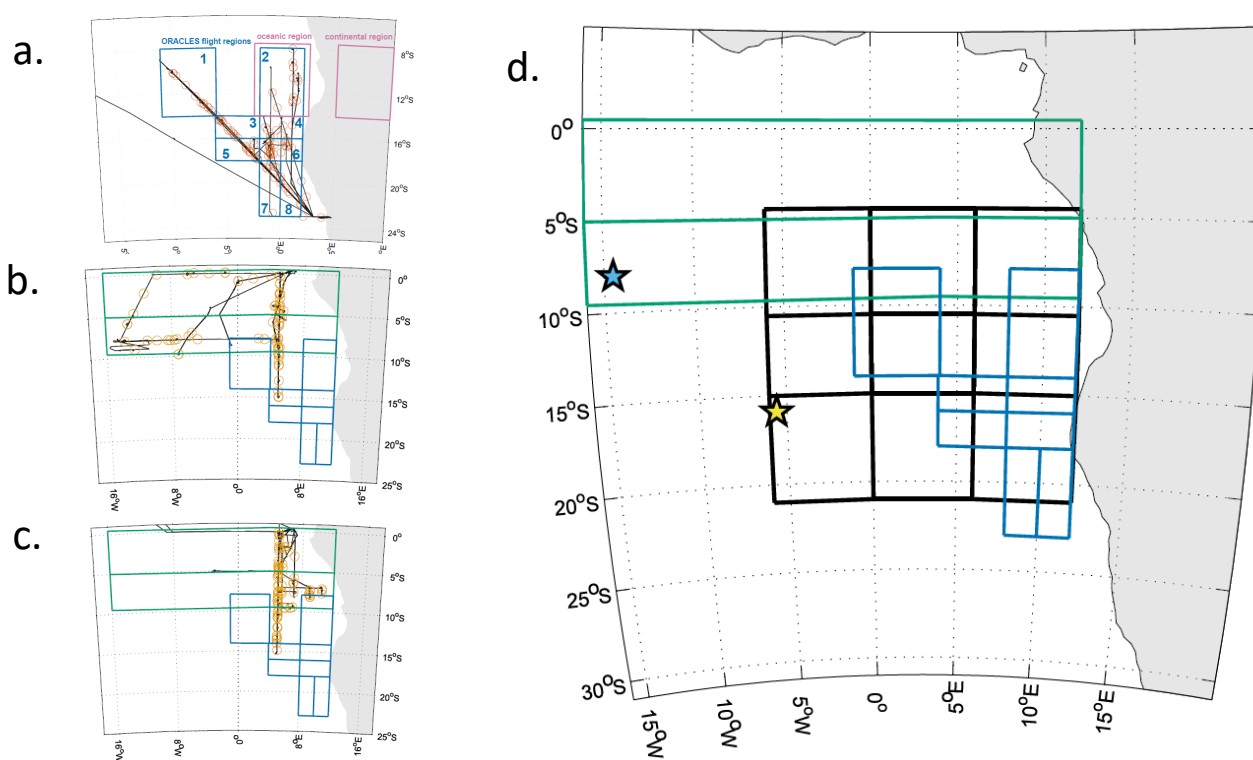

**Figure 1.** Regional distributions of observations and the spatial subdivisions used. (a) From Pistone et al. (2021), the locations of ORACLES-2016 aircraft profiles (orange circles), the regions used for the aircraft-based analysis (blue) and the reanalysis study (lavender). In (b) ORACLES-2017 and (c) ORACLES-2018, more northern profiles were sampled. For ease of comparison, these data (green grid) are divided at 5°S to aid comparison with 2016 (blue grid). Note also the larger westward range of observations in 2017. (d) Summary of the different subdivided regions considered in this and other works. The stars show the sites of previous island-based observations at St Helena Island (yellow star) and Ascension Island (blue star). The thicker black lines are the SEA divisions discussed in Section 4, on a nominally $6° \times 6°$ grid which is slightly uneven north-to-south to align with the CAMS spatial resolution and to isolate the AEJ-S latitudes.



## 3 Agreement between reanalyses and aircraft observations

In assessing the representativeness of the reanalyses and their agreement with the ORACLES observations, we begin with specific humidity $q$, a meteorological field, and CO, a biomass burning tracer. We choose CO as our BB tracer as the modeling of chemical species is subjected to fewer uncertainties than is aerosol modeling. Aerosol as modeled in the CAMS reanaly-

sis considers aerosol speciation, processing, lifetime, and removal for 12 separate aerosol components, while the assimilated observation is column total AOD from satellites (Inness et al., 2019). Because of this, we expect CO to have a more straightforward and accurate representation than the fields of aerosols themselves, for our purposes of assessing and characterizing atmospheric distributions over time. We note that the lifetime of CO in the atmosphere is 1-4 months (Szopa et al., 2021) and thus may show accumulation to a higher background value over a given biomass burning season (3 months), but as we show in

Section 3.1, the observed aerosol-CO relationship does not vary across the three airborne campaigns.

Previous work in Pistone et al. (2021) showed that the ERA5 reanalysis captured the vertical structure and location of the humid plume remarkably well in September 2016 (there reported as an observed/ERA5 correlation of $R^2 = 0.79$ for observations above 2km over all flights). ERA5 performed better than MERRA-2 in terms of this direct comparison (observed/MERRA-2 $R^2 = 0.40$), due to a known vertical velocity (excessive subsidence) issue in the latter (e.g., Das et al., 2017). Figure 2 shows

comparisons between (left, middle, right) ERA5, CAMS, and MERRA-2 versus the observed values from the 2016 ORACLES deployment for (top, middle, bottom) specific humidity, CO, and the relationship between the two. The CAMS reanalysis shows a similar pattern in specific humidity and agreement with the observations ($R^2 = 0.84$ for $z \geq 2$km), despite its lower spatial and temporal resolution (Section 2) giving roughly half as many matches compared with ERA5. In all the reanalyses considered, the largest reanalysis-observation discrepancies occur near the top of the boundary layer (∼1-2km).

For the CO comparisons (Figure 2, middle row), there is similar good agreement overall between CAMS and observations ($R^2 = 0.75$ for $z \geq 2$km) although the reanalysis tends to underestimate these values for higher observed CO ($\gtrsim 300$ppb). The overall slope between the CAMS and observed CO is 0.79, compared with 0.98 for the CAMS $q$ versus observed $q$. Still more variability is seen in the direct comparison with MERRA-2 (CO observed/MERRA-2 correlation: $R^2 = 0.21$). This general pattern consistent over all three ORACLES deployments (Figures 2, 3, and 4), and is also consistent with Pistone et al. (2021),

likely due to the aforementioned issue with vertical plume displacement. In other words, MERRA-2 tends to underestimate the higher altitude points while sometimes overestimating lower-altitude points for both CO and $q$ relative to aircraft observations, while still preserving the relationship between CO and $q$ for MERRA-2 overall (e.g., Figure 2, bottom right).

The data from August 2017 (Figure 3) and October 2018 (Figure 4) are largely consistent with the picture from September 2016, with a few notable differences. First, the correlations of observed versus CAMS or ERA5 water vapor are slightly better

than in 2016 and substantially better for MERRA-2 (Table 1, with the caveat that there are fewer profile matches for these years). However, the CO observed versus CAMS is a somewhat poorer match both in terms of the $R^2$ values and the slope for the latter deployments. This may be explained by the difference in dynamic range measured year to year following seasonal evolution of the smoke plume; specifically, higher CO observed in August 2017 results in a slope which is skewed low due to



**Table 1.** $R^2$ values, total number of matched observations, and total least squares linear fits for aircraft and reanalysis data. Data are shown for the subset matching the profiles in Figure 1, and are for altitudes 2km and above (i.e., 800-400hPa levels) and south of $5°$S for 2017 and 2018 (i.e., AEJ-S-influenced latitudes). The left columns show correlations between specific humidities, while the right side shows the correlations for CO. All R-values are statistically significant at $p < 0.0001$.

| data | obs $q$ vs $q$ from | | | obs CO vs CO from | |
|---|---|---|---|---|---|
| | ERA5 | CAMS | MERRA-2 | CAMS | MERRA-2 |
| Sep 2016 | | | | | |
| $R^2$ | 0.78 | 0.84 | 0.37 | 0.74 | 0.22 |
| num obs | 516 | 237 | 516 | 237 | 515 |
| fit | 1.07x+0.12 | 1.03x+0.19 | 1.28x+0.40 | 0.80x+15.7 | 1.02x-31.68 |
| Aug 2017 | | | | | |
| $R^2$ | 0.88 | 0.86 | 0.80 | 0.67 | 0.40 |
| num obs | 305 | 141 | 305 | 140 | 301 |
| fit | 1.006x+0.074 | 0.996x+0.14 | 1.10x+0.06 | 0.75x+7.78 | 1.002x-18.4 |
| Oct 2018 | | | | | |
| $R^2$ | 0.91 | 0.92 | 0.86 | 0.55 | 0.32 |
| num obs | 317 | 149 | 317 | 143 | 304 |
| fit | 0.97x-0.075 | 0.96x+0.21 | 0.95x+0.53 | 0.60x+62.1 | 0.70x-3.33 |

the CAMS high-CO underestimation, and the lower CO observed in October 2018 results in a less robust $R^2$ value due to a smaller dynamic range.

In addition to the direct observation/reanalysis comparisons, Figures 2 to 4 also show how the correlation between CO and $q$ is represented by each entity (bottom row). The differences between plume level (orange circles) correlations versus PBL observations (blue squares) are evident (i.e., humid but low-CO air masses are found only in the oceanic PBL), but overall these relationships continue to be well-represented, despite the mismatch in location for reanalyses versus observations (e.g. the displacement in MERRA-2). As was discussed briefly in Pistone et al. (2021), the magnitude of the observed CO-$q$ relationship varies across different deployments/months (from observations, slopes were 0.020, 0.023, and 0.05 (g/kg)/ppb for August, September, and October, respectively), due in large part to the changing meteorological and BB conditions, namely the increase in humidity and decrease in BB loading as the season progresses (Adebiyi et al., 2015; Ryoo et al., 2022).

In light of the better CAMS and ERA5 agreement with observations compared to MERRA-2, in the subsequent sections we focus our analysis on the ECMWF reanalyses (higher-resolution ERA5 for water vapor, and CAMS for chemistry).

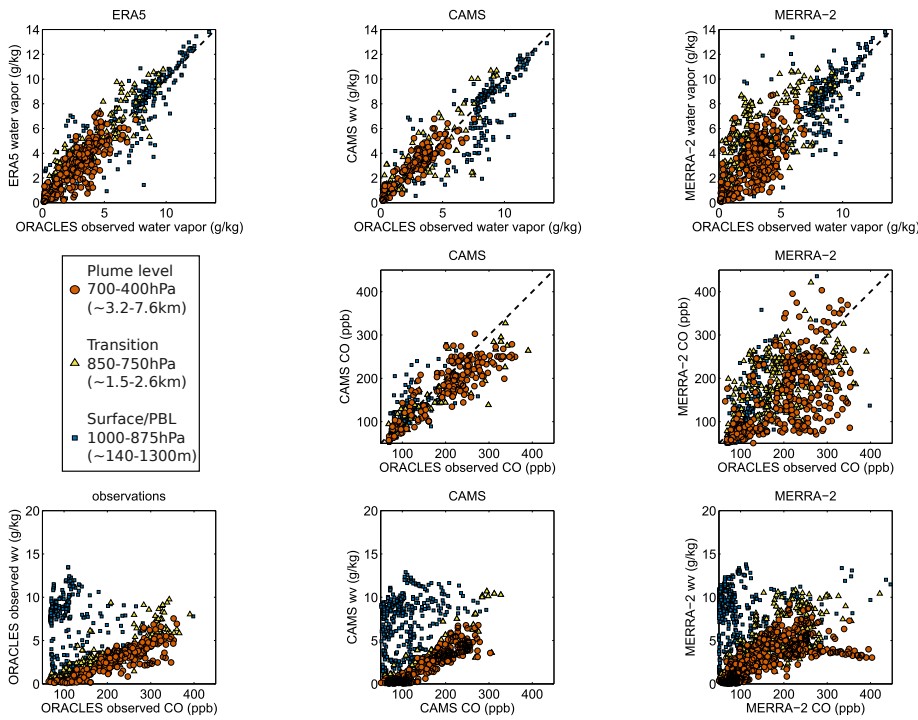

**Figure 2.** Scatters of water vapor and CO, for September 2016 and by altitude, for data collected during P-3 aircraft vertical profiles (circles in Figure 1a). Comparisons are shown for the (left) ERA5, (middle) CAMS, and (right) MERRA-2 reanalyses. Top row: observed water vapor (average observed over a 100m layer) compared to coincident reanalysis values. Middle row: observed CO compared with the same for CAMS (center) and MERRA-2 (left). Bottom row: collocated CO versus water vapor for each product (aircraft observations, CAMS, and MERRA-2). Blue squares= 1000-875hPa (surface to ∼1.3km, boundary layer); yellow triangles = 850-750hPa (∼ 1.5 − 2.6km, BL top/transition); orange circles = 700-450hPa (∼ 3.2 − 7.6km, BB plume). The 1:1 line is shown as black dashes. Correlation coefficients and slopes of the fits for data >2km are given in Table 1.

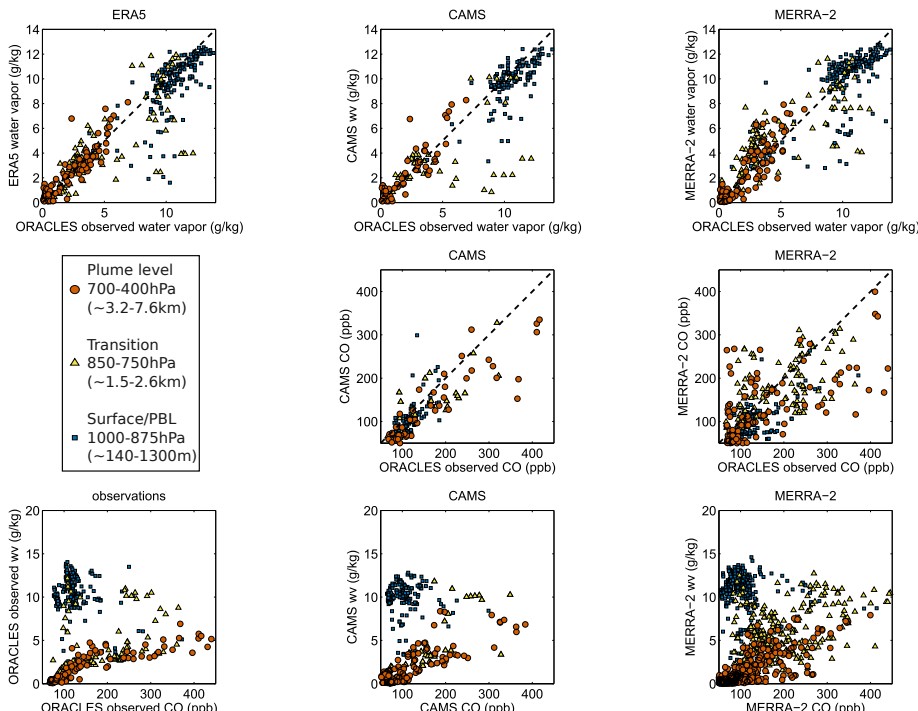

**Figure 3.** As in Figure 2, but for August 2017. Data shown here are from profiles south of 5°S (Figure 1b) to isolate the airmasses influenced by the AEJ-S, although the results are consistent for the larger dataset. The bottom left (observed CO versus water vapor) illustrates the greater range in CO values as compared with 2016.

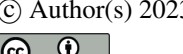



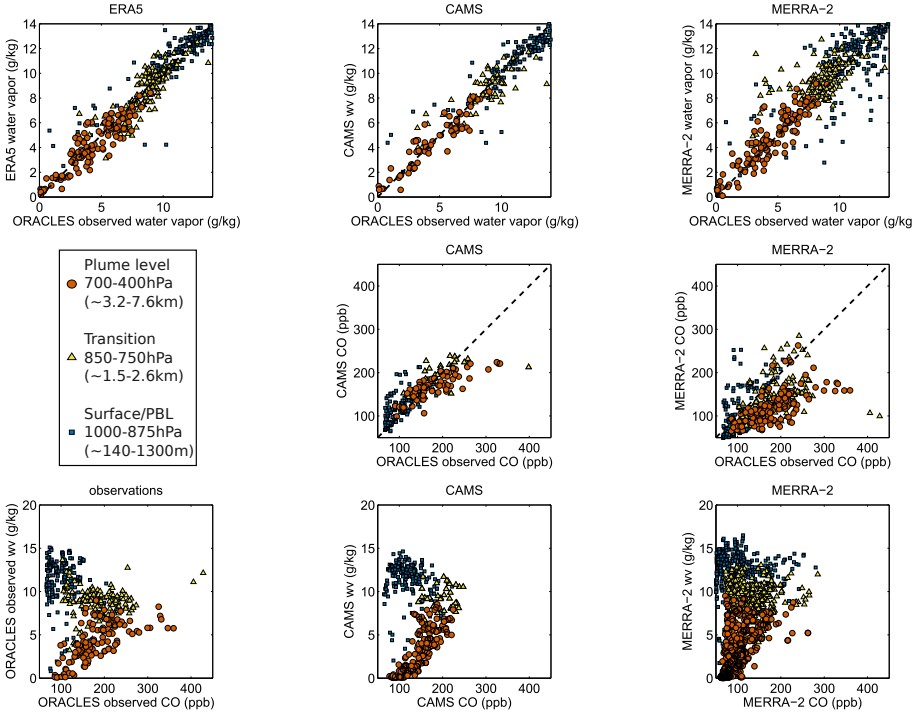

**Figure 4.** As in Figure 2, but for October 2018. Data shown here are from profiles south of $5°$S (Figure 1c), although the results are consistent for the larger dataset. Here, the most striking feature is the lower-CO, higher-$q$ conditions (i.e., steeper slope in the bottom row) as compared to observations from earlier in the BB season.



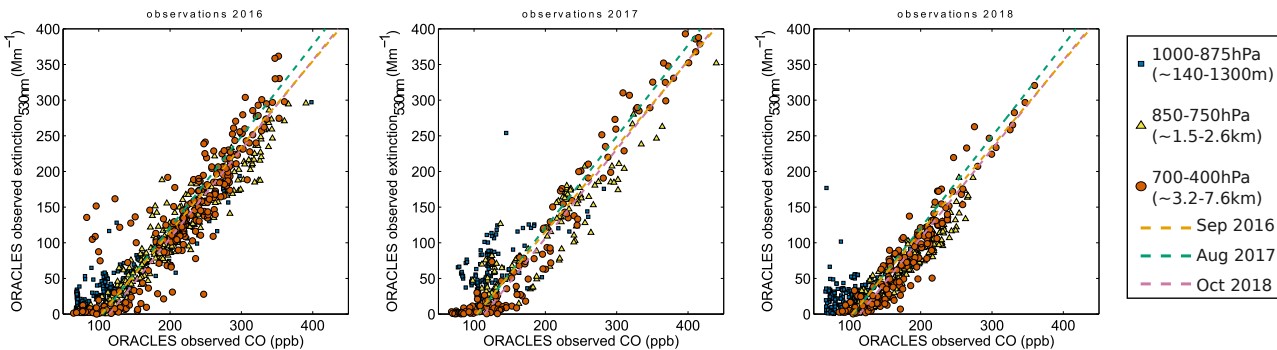

**Figure 5.** Inlet-based observed CO and aerosol extinction (in-situ scattering + absorption) for ORACLES deployment, for the same profiles shown in Figures 2-4. The $R^2$ correlations are >0.9 for plume level (>700hPa) for 2017 and 2018, and $R^2$=0.83 for 2016. The lower 2016 correlation was due to issues with dead air in the nephelometer which produced artifacts while profiling during that deployment. Lines show the total least-squares fits to all data above 3km for each year, and the relationship is consistent throughout the season (slopes of 1.20 to 1.28) despite variability in loading.

## 3.1 Aerosol loading and aerosol optical depth

Ultimately, our interest in characterizing the atmospheric structure in this region stems from a desire to understand and quantify the aerosol radiative and dynamic effects of the BB plume. We have focused on a BB tracer field from the reanalyses (i.e., CO) because, due to the complexities in atmospheric aerosol processing in the real world and in models, CO is a simpler parameter

to model accurately compared with speciated aerosol. Of course, this exercise will be only tangential to the radiative question if the CO tracer does not correspond to aerosol loading in the real world. To explore whether this is the case, Figure 5 shows the observed CO versus extinction (nephelometer scattering plus PSAP absorption; Section 2.1.3), subset to the same profiles as Figures 2 to 4. There are a few outlier points in 2016 (during this deployment, the nephelometer had an issue with dead air when entering or exiting the plume altitudes), but overall the correlations are strong ($R^2 > 0.9$ for 2017 and 2018, and 0.83 in

2016). Importantly, the slopes themselves are consistent year to year (dashed lines; total least-squares fit slopes of 1.20 to 1.28 over the three months) suggesting the CO-aerosol extinction relationship does not exhibit a significant seasonal dependence, at least for these dried, inlet-based measurements.

Figure 6 shows comparisons between 4STAR-measured AODs and CAMS AOD (both at 550nm) for the same profile locations. Note that we show both reported "total" CAMS AOD as well as the sum of the CAMS-reported organics and BC

components of AOD, to isolate the BB plume signal from potential other sources such as sea salt. Sea salt in particular is





likely to be excluded from much of the ORACLES AOD measurements, which only include above-aircraft column AOD due to 4STAR viewing geometry.

Figure 6 shows the variation in correlations and spreads for the different deployments. Contrasting the years, we see a poorer correlation and greater spread in CAMS-4STAR differences in September 2016 compared to later years (direct differences are
shown in Supplementary Figure S1). Table 2 summarizes the correlations, slopes, and agreement between the observations and the CAMS BB AOD. All years share some features: for all years, using only BB aerosol types in CAMS brings the average more in line with the observations compared with "total" AOD, but this offset is reasonably consistent across all data. Also, in all years, particularly for higher loading conditions, CAMS tends to overestimate AOD relative to 4STAR, a curious feature since, as shown previously, CAMS tends to underestimate CO concentration particularly for higher loadings (Figs 2 to 4).
As noted in Section 2.2, additional the satellite-observed aerosol is total-column AOD, and CAMS aerosol is then partitioned into a speciated model, with aerosol and chemistry largely independent of one another (Inness et al., 2019). Because of this, and previous results showing that total AOD from MODIS in this region is greater than that observed from aircraft due to the difficulties in retrieving aerosol above cloud (LeBlanc et al., 2020), it is not surprising that CAMS overestimates AOD relative to the aircraft observations, as the CAMS AOD is a less constrained parameter.

Regardless, the majority of the available comparisons agree within AOD$\pm0.2$ of one another (Figure S1; Table 2), although CAMS tends to overestimate AOD relative to 4STAR in all years, even when only BB AOD is considered. We note that Cochrane et al. (2022) found that aerosol heating rate increased by $\sim 0.5$ K/day per 0.1 increase in AOD, suggesting that it is worth further characterizing the nature and impacts of a disagreement of this magnitude. Potential sources of uncertainty from the 4STAR measurements include only partial columns due to aircraft altitude; indeed it is particularly evident in the
2016 and 2017 panels of Figure 6 that the higher altitudes (yellow points, i.e., a smaller fraction of a "full column") tend to show the largest negative deviations compared with CAMS, possibly due to 4STAR missing below-aircraft AOD contributions. Differences in spatial sampling may also contribute to the year-to-year differences.

In terms of column aerosol measurements, Pistone et al. (2019) demonstrated that 4STAR AODs, while generally consistent with other measurements of optical depth, tended to be slightly larger than co-located values by other measurement methods
(there, integrated column extinction from inlet-based instruments and remotely-sensed measurements from downward-looking sensors), likely due to the differences in viewing geometry. Chang et al. (2021) showed good agreement both between the 4STAR AOD and HSRL-2 AOD products, and between 4STAR and several satellite-derived (MODIS and SEVIRI) ACAOD products, for conditions of close temporal collocations. Under more relaxed temporal agreement constraints, when compared just with two different MODIS ACAOD algorithms, 4STAR tended to report slightly lower ACAOD than one, and slightly
higher than the other. The authors concluded that these results suggested aerosol loading conditions over the SEA can vary substantially within a 3-hour time period, which may also help to explain the poorer agreement of the 3-hourly CAMS reanalysis relative to the hourly ERA5 results for $q$, although the AOD results in Figure 6 are not altered by excluding profiles with the greatest temporal mismatch between reanalysis and flights.

Interestingly, when we consider CAMS column CO versus column BB (organics + BC) AOD, the relationship does vary
with season (example shown in Supplementary Figure S2). For each of the deployment years 2016-2018, the October data





**Table 2.** Agreement between CAMS AOD (organic + BC) and 4STAR ACAOD for the profiles shown in Figure 1 from each ORACLES deployment. All correlation coefficients are significant with p<0.0001.

| year | 2016 | 2017 | 2018 |
|------|------|------|------|
| fit | 1.08x+0.002 | 0.89x+0.10 | 1.43x-0.09 |
| $R^2$ | 0.34 | 0.64 | 0.74 |
| $R$ | 0.59 | 0.80 | 0.86 |
| # points | 73 | 41 | 39 |
| $|\Delta AOD|<0.1$ | 52% | 66% | 79% |
| $|\Delta AOD|<0.2$ | 84% | 88% | 97% |

show a steeper slope (i.e., more AOD per unit CO) than do the earlier months, similar to the pattern shown in Figure 2 vs Figure 4. This is not inconsistent with Figure 6, considering the contributions of the boundary layer to modeled column values, and this could potentially be affected by aerosol hygroscopicity, which is included in the CAMS aerosol scheme (Inness et al., 2019; Morcrette et al., 2009) and which would likely be more prominent in the more humid October months. We note that

5    Shinozuka et al. (2020) concluded that $90\%$ of free-tropospheric ORACLES-2016 measurements were not significantly affected by hygroscopic swelling ($f(RH) < 1.2$) although this effect was more pronounced in PBL measurements ($f(RH) > 2.2$ for half the valid inlet-based measurements). Hygroscopic swelling would not be a factor in the inlet-based aircraft measurements (Figure 5) which have been dried to RH$< 40\%$.These considerations complicate studies using modeled or observed aerosol extinction, which explains our approach of using a CO tracer while also emphasizing the importance of field measurements in

10    the region.



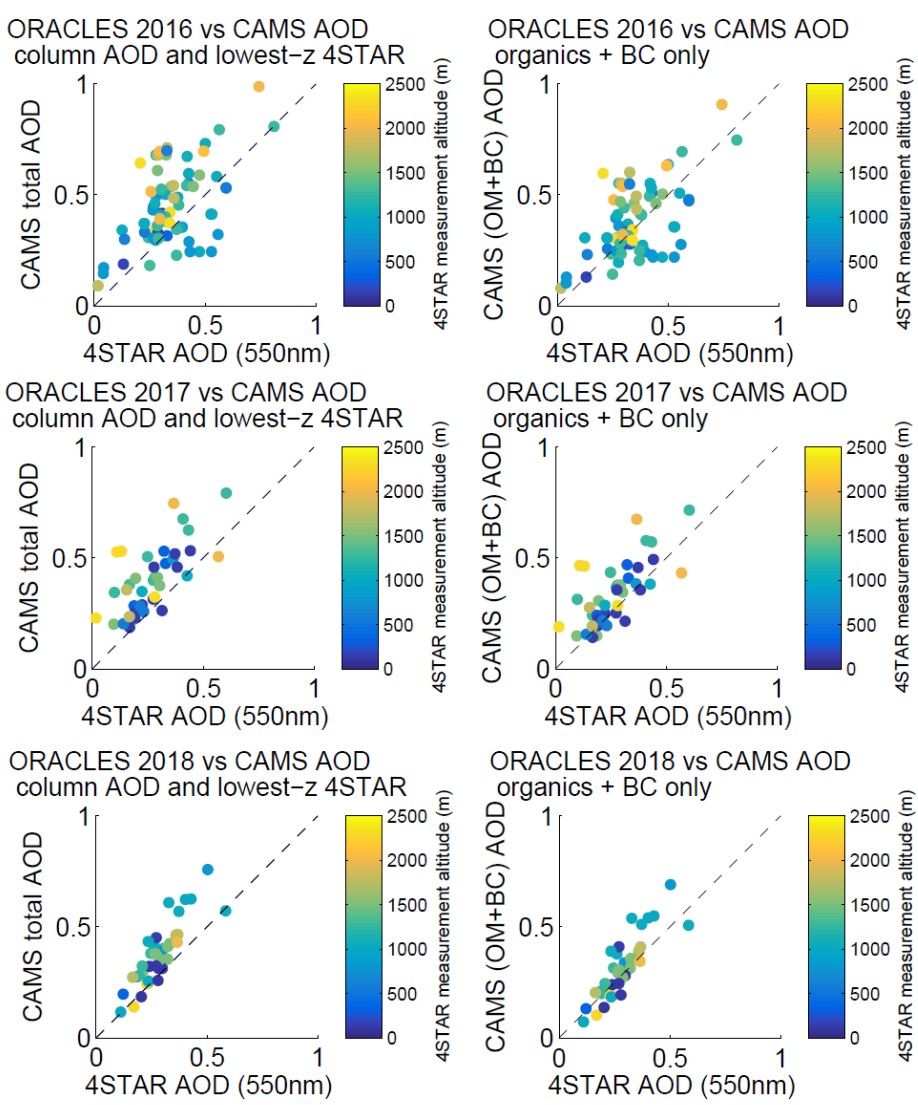

**Figure 6.** 4STAR to CAMS, for 2016 (top), 2017 (middle), and 2018 (bottom) deployments at the location of aircraft profiles (Figure 1) for total (left) and BB-only (right) CAMS AODs. Colors indicate the altitude of the aircraft measurement (i.e., the lowest value in the given aircraft profile); the observed AOD is for the column above this altitude, compared to the full-column values reported by CAMS. Dashed lines show the 1:1 line.



## 4 Spatial and temporal incidence of atmospheric vertical profiles: k-means analysis

Despite the considerations discussed above, our analysis shows a general good agreement between the ORACLES aircraft data and the ERA5 and CAMS reanalyses from all three deployments. These products are thus useful to the end goal of a more complete characterization of the atmospheric structure of the region as a whole. We approach this section with the goal of being

able to characterize the frequency of profiles over the SEA as a whole and climatologically during the BB season, not solely during flight periods and locations.

To accomplish this, we performed a k-means cluster analysis, similar to the methods used by Schelbergen et al. (2020), on the full reanalysis subset of 3-hourly August, September, and October data from 2014 through 2020, from the equator to 30°S and 15°W to 12.5°E (i.e., near to the coast without including land). We use vertical pressure levels between 1000hPa to

400hPa, giving 18 levels of $q$ from ERA5, and 10 levels of CO from CAMS. First we perform a principal component analysis separately for $q$ and CO by decomposing the full set of all profiles for each variable into two principal components describing the variability of the profile (principal components shown in Supplementary Figures S3 and S4). Then the distribution of ERA5 profiles in the phase space of the two principal components is passed to a k-means clustering algorithm to arrive at a set of canonical profiles for each variable. For each variable, six clusters (shown in Figures 7 and 8) were determined from the two

principal components, whose variability corresponded to (1) total concentration (i.e., profile q1 vs q2 vs q3) and (2) the ratio of upper-level ($\sim 600 - 700$hPa) to lower-level ($\sim 900$hPa) values (i.e., q1 vs q4 vs q6; distributions across the two principal components are shown in Supplementary Figures S5 through S8). The profile labeling corresponds roughly to the column concentration, i.e., q1 is the most humid and q6 is the driest; C1 has the highest CO concentrations and C6 has the least. We chose a 6-cluster configuration as with this number we were able to adequately capture the variability of conditions (described

below), while limiting the number of defined profiles for ease of interpretation. ERA5 outputs subsampled to the CAMS spatial resolution produce similar results, and the k-means cluster method produces similar results (i.e., the same general types and range of profile types) for slight differences in the spatial domain of the data used in the analysis and for the number of clusters chosen. A silhouette analysis of various configurations between 3 to 9 clusters suggests the addition of more clusters is no more optimal compared with the 6-cluster configuration.

The identified profiles span the range in conditions seen spatially and through the BB season. For water vapor (Figure 7), there is a range in both total concentration and vertical structure across the six profile types. In terms of water vapor concentration, the range in surface $q$ varies from 7.8 g/kg to 14.9 g/kg, with q6 and q5 being the driest profiles and q1 and q2 being the most humid. At the plume level ($\sim$700 hPa), $q$ ranges from 7.2 to 1.0 g/kg across the 6 profiles, and all profiles approach 0 above 500hPa (0.2 to 0.5 g/kg at 400hPa). Profiles q5 and q6 are also similar in that they both have a dry free

troposphere ($> 800$hPa, or $\sim 2$km); the principal differences distinguishing the two are a more humid/slightly deeper BL in q5, and the presence of a dry air gap in q6, with slightly greater humidity at 600-700hPa compared with at 800hPa.

Regarding vertical structure, two of the profiles (q1 and q2) are structurally similar, with a very humid PBL (with $\sim 2$g/kg difference in surface $q$) below a constant monotonic decrease in $q$ with altitude. Two other profiles (q6 and q4) feature a dry gap at the top of the PBL with increased humidity above ($600 - 800$hPa), with 0 being the driest profile overall; again the surface



**Figure 7.** (left) The six k-means profile types determined from ERA5 reanalysis water vapor. (right) Fractional incidence of k-means water vapor classifications, by month, for ERA5 August-October 2014-2020. The grid corresponds to the black outlines in Figure 1d. Note that the top and middle rows correspond to the AEJ-S latitudes (5°S-15°S), with the top row slightly extended to align with the CAMS 0.75° resolution used in the following Figure.

$q$ difference is slightly less than 2g/kg between the two. The final two (q3 and q5) are intermediate; they show a well-defined boundary layer and a layer of more uniform humidity above. The remainder of Figure 7 shows how the distribution of these profiles varies over the region.

There are notable spatial and temporal variations in the incidence of the profile types. Figure 7 shows the fractional incidence of each water vapor profile within the gridded SEA region (Figure 1d), for each month. Given the continental source of the





**Figure 8.** (left) The six k-means profile types determined from CAMS reanalysis CO. (right) Fractional incidence of k-means CO classifications, by month, for CAMS 2014-2020 for the regions shown on the black grid shown in Figure 1d.



water vapor and the seasonal evolution seen in the aircraft data, it is not surprising that in the northeast of the SEA basin (4.5-10.5°S, 6-12°E), the most humid profiles (q1 and q2) are most frequent. The most humid, q1, increases in frequency as the season progresses, from 26% of August profiles within this domain to 54% in September and 72% of October profiles (50% overall). Considering the sum of the two most humid profiles (q1 and q2), 70% in August and 94% in October (85% overall) of

the profiles fit this classification. This is consistent with continental influence in these domains, as well as warmer sea surface temperatures in the northeast compared to the rest of the region (e.g., Zuidema et al., 2016).

The northern sectors farther from the coast (10.5-15°S, 0-6°E and 6°W-0°E), also have a high incidence of the high humidity profiles (q1 and q2), with the frequency dropping off with distance from the continent (i.e., 21% q1 and 43% q2 overall in the middle region and 9% q1 and 42% q2 in the west-most region shown). This is again consistent with a pattern of continental

outflow of humid air which experiences a gradual mixing with less-humid airmasses as it moves over the SEA basin. There are also significant differences in air mass transport based on altitude, due to the presence of the AEJ-S in this region, with a maximum in the northeast sector and 600-700hPa (Figure 9). The AEJ-S weakens further from the coast, and a northerly component of the wind interrupts the direct-westward transport and brings the humid outflow to higher latitudes, which may be responsible for the high incidence of profile q4 in the middle of our study region.

Further south (Figure 7, bottom row), the atmosphere is significantly less humid. In the southwest sector (15-21°S, 6°W-0°E), q5 is dominant, making up 52% of profiles over the three months (72% of August, 54% of September, and 32% of October profiles). This condition is most common in August and far-from-coast, likely due to dry air intrusion from the southwest (Ryoo et al., 2021) and more limited contribution of airmasses from the continental source as in much of the rest of the Southeast Atlantic. South of the AEJ-S latitudes, profile q4 is also frequent (28-45% over all months) and the q6 profile is increasingly

common (36-40% in the bottom center and bottom east boxes: 15-21°S, 0°E-12°E).

In this visualization of water vapor profiles shown in Figure 7, one can see two different air masses sourced into this region: the humid, monotonically-decreasing airmasses from the northeast and moving south and west in the AEJ-S (q1, q2), and the dry, more uniform airmasses above the boundary layer in the southwest, moving north and east (q5, q6), and the middle row of Figure 7 is where they meet. In the poleward half of the AEJ-S latitudes (10.5-15°S, 6-12°E; middle row), q4 is the predominant

profile type in the middle region (10.5-15°S, 0-6°E): 54% averaged, and fairly consistent month to month with 46% in August, 53% in September, and 63% in October. This profile with an elevated humidity layer of $\sim 5$g/kg around $\sim 650-800$hPa, and a drier gap above the (again more humid) boundary layer likely results when an elevated humidity air parcel moves with AEJ-S velocity while being subjected to vertical subsidence, whereas drier air at lower altitudes originates from another, non-continental source. Profile q4 is also common over much of the southernmost part of the region (28-41% south of 10.5°S), and

increases throughout the season. The fact that a less-humid gap is present between free tropospheric humidity and boundary layer humidity highlights the air mass sourcing as over the continent at high altitude. Overall, this means that for these higher latitude regions (south of 10.5°S), the jet-altitude air appears to more frequently originate from high altitude in the north rather than from the continent directly to the east. This anticyclonic transport was also observed in the trajectory analysis in Pistone et al. (2021) and illustrated in Redemann et al. (2021); Ryoo et al. (2021).





**Figure 9.** Monthly-averaged zonal winds at 600hPa (left) and 700hPa (middle), corresponding to the AEJ-S altitudes for September/October and August, respectively. The easterly transport north of $10 - 15°$S is evident. (Right) Monthly-averaged profiles of wind (u,v) at the ERA5 pressure altitudes for August (blue), September (orange), and October (yellow), in the same grid. Dashed horizontal lines show the altitude subdivisions used in Figs 2 to 4. The northerly motion is evident at the lower altitudes (below 700hPa).



Profile q3, of intermediate total humidity with a humidity gradient at the PBL top, may be seen as a less-humid evolution of q1 or q2, likely resulting from mixing of continental air above $\sim 850$hPa ($\sim 1.5$km) with drier airmasses as the continental air moves farther from the its source (as with q4). Within this middle latitude range, q3 is a consistently significant though not dominant fraction of the profiles (10-36%), with prevalence increasing with distance from the coast and decreasing through the season (August to October) as overall humidity increases.

While profiles q5 and q6 are uncommon north of $15°$S (with the exception of the western sector, with 43% q5 in August, 25% overall), and similarly profile q1 is much less common south of $10.5°$S, comprising only 6% of the total profiles, and those mostly in October (1% August, 3% September, and 14% October). An additional quarter of profiles are classed as q2 in this near-coast region. Profile q1 is almost entirely absent further from the coast (<2% in 0-6°E) at this latitude range; these patterns are all reasonable with the climatological transport described above. Considering only this water vapor picture, the story is fairly straightforward, but the biomass burning tracer adds more complexity to the picture.

The conditions captured in the 6 k-means-defined CO profiles (Figure 8) are somewhat less varied in structure than those for $q$. While there is still variability in BL CO (66 to 950ppb), the primary difference is the range at in the free troposphere, from high concentration ($\sim 400$ppb) to a background value of 60-70ppb. The peak concentration is seen either at 700 hPa (C3, C5) or 800 hPa (C1, C4), with C2 nearly uniform across this range ($\sim 2 - 3.2$km). This is lower in altitude than the typical maximum of the south African Easterly Jet (AEJ-S) which is at 600-700hPa, (Adebiyi and Zuidema, 2016; Pistone et al., 2021; Ryoo et al., 2022); the coarse pressure level resolution of the CAMS reanalysis may be a factor, plus there is a degree of vertical subsidence with time over the region ($\sim 50 - 80$hPa/day) which is seen in both the reanalysis and the observations, as was previously shown.

We note that curious outcome of applying this method to CO profiles is that profile C6 (the background/smokeless case) shows slightly increasing CO with increasing altitude, from 65ppb at 1000hPa to 88ppb at 400hPa. We suspect this may not be a real characteristic of the atmospheric structure, but rather may be an artifact resulting from limitations in the CAMS satellite assimilation. Previous studies have documented low biases in CO in the lower and middle troposphere when compared with aircraft profiles (e.g. Inness et al., 2019, 2022). Specifically, Inness et al. (2019) showed consistent low-biases of 10-20% in CAMS reanalysis CO between 600hPa and the surface, for all airport sites considered. While the Windhoek, Namibia (i.e., closest geographic to our study area) site showed consistent discrepancies through these altitudes with only a slight decrease in magnitude near the surface, other sites show a pronounced increase in the negative bias at the lower altitudes relative to higher in the troposphere. The complicated atmospheric structure in this region makes it plausible that this may also be occurring in the present case. In other words, profile C6 is not showing a true decrease in CO with $z$, but rather an unphysical artifact in an actually-constant-CO atmospheric structure. Manual inspection of some C6 profiles indicate the presence of a small CO layer in some cases, and a more uniform profile in others, suggesting this may be a limitation of our k-means classification method, which causes such less-smoky profiles to be collapsed into one case. The minimum CO observed (i.e., background) by aircraft during ORACLES is around 60ppb, closer to the low-altitude values in this profile, which suggests instead a potential overestimation in CO at higher altitudes rather than an underestimation at the lower altitudes. Nonetheless, the presence at times of an effectively smoke-free atmospheric profile is consistent with expectations in this region. We will leave the specific



implications of the particular shape and incidence of C6 for a future work. Overall, this k-means method produces a range of profile types which are consistent with the observations across the SEA.

The CO profile classifications (Figure 8) are in many ways more straightforward than those of $q$, but their spatial distribution is more complex. Compared with the $q$ distributions in Figure 7, the CO distributions are somewhat less continuous throughout the season, although some general patterns are still clear. The northeasternmost sector (4.5-10.5°S, 6-12°E) has the highest percentage of high-CO profiles, although the incidence of these profiles (C1 and C2, respectively) shows a seasonal trend opposite to that of $q$. Specifically, CO concentration is greatest in August and decreases in October, contrasting the increase in $q$ over this time. The profile frequencies for that sector are 41% (C1 only) and 85% (C1 or C2) in August; in September, C1 alone drops to 12% frequency, but the summed frequency of C1 plus C2 is still 73% overall. By October, less than 0.5% of profiles are classed as C1, and only around 19% are classed as C2. In other words, even as the water vapor increases from August to October, the CO (and by proxy, smoke) decreases dramatically, providing an opportunity to better examine the impacts of the dynamic range of the two profiles together.

There are other features which distinguish the distribution of CO profiles from that of the q profiles. There is a notable lack of a consistent trend across a three-month season within a specific region, particularly south of 10°S, in contrast to the consistent increase in $q$ (and increase in associated $q$ profile types) in all regions. Instead, the CO profiles often peak (or trough) in September, with lower (or higher) values in August and October (e.g., C3 in the northern and middle latitudes far from coast, and C2 near coast); alternatively, in other sectors the CO values increase (e.g., C5, south of 10.5°S) or decrease (e.g., C6, in the south) through the season. CO profiles in the middle and southern sectors are more likely to be classed as one of the more smoky profiles in September than in August or October, despite the progression of more- to less-smoky in the more northern regions. Some expected seasonal patterns are still evident; despite significant frequency of smoky profiles especially in September, by October, the middle-west region is dominated by the cleaner profiles (72% here are C6 or C5, i.e. one of the two least smoky profiles), and the northwestern and middle regions are almost half (45% and 48% respectively) C5 or C6. This reflects the decreased amount of smoke available for circulation later in the season even as the humidity profiles become more frequent in these same regions at this time (Figure 7) and monthly-average jet strength persists (Figure 9). This does not mean that this more remote region is not influenced by the biomass burning plume (indeed, this sector contains St Helena Island, whose smoke and humidity vertical structure are studied by Adebiyi et al. (2015) and others, and the Ascension Island observatory is located to the west of the northwesternmost sector we consider, i.e. even more removed from the source); but the smoke there is more episodic. By September, almost all profiles north of 15°S have some CO between 500-900hPa, while south of 15°S, 12-35% of profiles are still classified as "unpolluted" (C6), again likely due to intrusion of southwesterly airmasses (Ryoo et al., 2021). This variability indicates the importance of the combined effects of atmospheric circulation patterns over the region (Figure 9), the accumulation of CO due to its longer atmospheric lifetime, and the seasonal shifts in both fuel type and fire location through the season (e.g., Che et al., 2022).

Viewing the SEA atmospheric structure in this way gives us 36 different potential smoke/water vapor combined structures, but fortunately there are fewer in reality. Figures 10, 11, and 12 show the frequency of CO-$q$ profile combinations for each month divided into the same latitude/longitude sectors. Again, profiles are ordered from nominally least to most CO and $q$ on



their respective axes. The results are consistent with those shown in Section 3 using data from from the ORACLES flights: while there is a range of combinations in all months, the highest frequency combinations fall along the low-low/high-high continuum. The patterns of Figures 7 and 8 hold as well: the southernmost zones have the greatest incidence of clean and dry profiles from the southwest. The high-high conditions (q2/C2) are seen to be most frequent mid-season in September, and the

latitudes of the AEJ-S have the greatest CO and vapor, decreasing with distance from the coast and with time.

There is a good deal of spatial and temporal variation to how these profiles covary. For example, in August (Figure 10), moving from east to west along the northernmost sectors (following the AEJ-S outflow), there is a relatively disperse range of conditions that are generally high-q, high-CO combinations at around 10% each (i.e., medium/high: q2/[C3,C4], q3/[C2,C3,C4]), although the q2/C2 condition is still a frequent combination especially nearest to the coast (∼20%). In the middle latitudes,

conditions become progressively more distributed along the high-high/low-low continuum, q5/C5, q4/C5, q4/C4, again on the order of 10% each. The maximum frequencies in this range are q5/C6 (with a maximum of 24% farthest from coast) and q4/C3 (21% middle-longitudes). To the south, the low-CO, low-q conditions (q5/C6, q6,C6) dominate more significantly (a combined 70%, 59%, and 33% moving west to east), although episodes of smoky air still occur (27% C4 nearest to the coast), under a range of humidity conditions (for C4, the conditions are fairly evenly split between q4 and q6).

In September (Figure 11), the profiles more strongly favor more frequent incidence of fewer combinations in all regions (e.g., q1/C2, q2/C2 are 56% of cases in the northeast, and q5/C6, q5/C5 are 51% of cases in the southwest), with greater variation in $q$ profile type than in CO type. By October (Figure 12), when the highest-CO (C1) profiles are almost entirely absent, the most common northeastern coastal profile is high humidity, medium-low CO (i.e., q1/[C2,C3,C4] at 39%, 12% and 12% for a total of 64% of cases in this range), consistent with the aircraft observations. A good example of the seasonal

progression is in the middle-easternmost box, which shows a range of combinations in all months, but progresses from skewed high-CO, low-q (i.e., q5/C1 or q6/C4) in August to high-q, low-CO (q4/C5, q2/C4) in October, with around 10% incidence of each. This presentation also highlights how smokeless conditions are rarely seen after August at the northern AEJ-S latitudes, despite lower smoke emissions in October. We also see an evolution of the overall variability in conditions, with some sectors (e.g., September/October in the south) showing more variability in $q$ than in CO, while others show more variability in CO

than in $q$ (e.g., August in middle latitudes or October in the northern latitudes). We also note that while clear diurnal cycles in temperature, potential temperature, humidity, and CO, are evident in the reanalysis over land, these effects are muted once the air is transported over the ocean regions, and there are no significant changes in profile class across the diurnal cycle. The major variations in atmospheric structure seen here are spatial and seasonal, and may reflect the episodic nature of smoke transport in the AEJ-S (e.g., Ryoo et al., 2021; Pistone et al., 2021). While a bit daunting, this large range of conditions within

a limited spatial area offers an ideal framework in which to quantify the radiative impacts of vapor, aerosol, and their various combinations.







**Figure 10.** k-means classifications $q$ vs CO, for August. The profiles are ordered from left to right, clearest to smokiest, and from bottom to top, driest to most humid, and are standardized to show the fractional incidence of each profile in the same shading scale for each region. As described in the text, the spatial patterns of the smoke and humidity profiles are evident.





**Figure 11.** As in Figure 10, k-means classifications $q$ vs CO, for September.





**Figure 12.** As in Figure 10, k-means classifications $q$ vs CO, for October.



## 5  Discussion

Several previous studies highlighted the importance of examining the aerosol-water vapor covariance over this SEA region, and this work is relevant to those results. As was discussed earlier, Adebiyi et al. (2015) used a few idealized case studies (low, medium, and high AOD) based on satellite and radiosonde data at St Helena Island (15.9°S, 5.6°W) to estimate a maximum

increase in shortwave heating due to moisture of 0.12 K/day, compared to 1.2 K/day of aerosol SW heating. In the longwave, they found the presence of moisture resulted in a maximum net cooling of 0.45 K/day at the top of the moisture layer, with a consequent reduction in radiation passing through the humid layer into the boundary layer. Within the context of this current study, we expect these conditions, which fall into the southwest grid box of our framework, to reflect only a small amount of the total radiative forcing, as the higher plume concentrations are more frequent elsewhere over the SEA.

Deaconu et al. (2019) similarly noted the water vapor and aerosol covariance in their study using POLDER, MODIS, CALIPSO, and ERA-Interim, but this study focused on the less-humid June-August time frame and specifically were interested in the impacts of this layer on underlying clouds. They found roughly 6 K/day aerosol warming at the aerosol layer for defined high vs low aerosol cases, and a corresponding water vapor effect of 0.76 K/day SW (warming) and -0.14 K/day LW (cooling). We note that both this work and that of Adebiyi et al. (2015) chose a framework of essentially "low" or "high" aerosol, rather

than a more discretized classification, or one that considers the varying vertical structure of the atmosphere over this region.

A more targeted study was performed by Cochrane et al. (2022), who used high-vertical-resolution aircraft data to quantify the aerosol and water vapor heading rates for specific case studies from flights in ORACLES-2016 and -2017. The authors found that for the cases examined, the average maximum aerosol heating rate was 4.6 K/day and that of water vapor was 2.8 K/day, or $\sim 60\%$ of the aerosol value and 4-10 times greater than the water vapor heating of the other studies. The cases considered

in Cochrane et al. (2022) are noted to have had much higher aerosol loading (AODs typically exceeding 0.4, compared to the $> 0.2$ threshold used by Adebiyi et al., 2015). The localized nature of each of these previous results highlights the importance of being able to accurately characterize atmospheric conditions over different parts of a given region. We also note that the case studies of Cochrane et al. (2022) are included in the profile-based analysis we present here, which will allow us to directly assess the agreement between the approaches.

Another recent study from Johnson and Haywood (2023) calculated that BC absorption in models facilitated self-lofting (increase in buoyancy due to aerosol heating), which has various potentially significant impacts on atmospheric dynamics and aerosol lifetime. Over the SEA, they calculated an additional lofting of 0.5km, which may be significant given the consistent subsidence in the region, delaying the eventual mixing into the underlying cloud deck.

Here, we have utilized observational datasets to describe the atmospheric structure of the SEA during the springtime BB

season, and to assess how well the unique vertical features of the region are captured in several frequently-used reanalyses. Using different elements of the comprehensive ORACLES airborne dataset, we conclude that the ECMWF reanalyses offer a fairly accurate characterization throughout the season August-October, and that the vertical structure of the CO tracer is a reasonable proxy for aerosol optical effects. These results are important because the radiative conditions vary over the region and with the season. The radiative heating of a given atmospheric layer strongly depends on local insolation/solar zenith





angle (e.g. Cochrane et al., 2022) which varies spatially and seasonally. Additionally, by characterizing the spatial (specifically longitudinal) variation of conditions in the region, we may be able to gain a better understanding of the lifetime influence of a particular airmass, i.e., the temporal extent of its radiative influence on the atmosphere/cloud column. This framework allows us to characterize the conditions in a more digestible format, and to construct a more realistic atmospheric frequency distribution.

The results of this study can aid in performing radiative transfer calculations in the regions of plume maximum, allowing us to put both isolated aircraft-based case studies and analysis of surface-based measurement sites into broader context. This will be the subject of a future work. Overall, the large range of conditions seen here, validated by the ORACLES aircraft measurements, offers an intriguing opportunity to explore the differing radiative impacts of water vapor in conjunction with a radiatively active BB plume.

**6   Conclusions**

Here, we have explored various aspects of the observed BB layer and co-incident elevated humidity signal over the SEA. We show the observed CO-$q$ relationship is consistent across three field deployments across the BB season, although the magnitude of the relationship varies (slopes of 0.020, 0.023, and 0.05 (g/kg)/ppb for August 2017, September 2016, and October 2018, respectively), due to changing meteorological and BB conditions as the season progresses. Despite the differing locations and

timing of each deployment, we have shown that good agreement between the airborne ORACLES dataset and large-scale reanalyses is consistent across the deployments, specifically for the ECMWF ERA5 and CAMS reanalyses.

Consistent with the results of Pistone et al. (2021) for September 2016, ERA5 reanalysis of water vapor agrees well with the ORACLES observations ($R^2 = 0.78, 0.88,$ and $0.91$ for September 2016, August 2017, and October 2018, respectively). The CAMS reanalysis also agrees well with observed water vapor ($R^2 = 0.84, 0.86,$ and $0.92$ for September 2016, August 2017, and

October 2018, respectively), albeit with lower resolution than the ERA5 reanalysis. CAMS represents water vapor somewhat more realistically than it does CO (campaign-wide $R^2 = 0.37, 0.80,$ and $0.86$ for September 2016, August 2017, and October 2018), with CO tending to be underestimated relative to observations especially under high smoke concentrations. In contrast, CAMS column aerosol optical depth (AOD) shows similar correlations to those of CO, but is generally overestimated relative to the field measurements. While NASA's MERRA-2 reanalysis preserves the observed correlation between CO and q, the smoky

airmasses are frequently displaced (too low in altitude) in MERRA-2 relative to the observations. This is consistent with the results of Pistone et al. (2021), which focused on the September 2016 observations. Nonetheless, inlet-based observations show a consistent relationship between CO and aerosol extinction across all observations, demonstrating the utility of the CO tracer to understanding vertical aerosol distribution.

Following this good agreement between ORACLES and CAMS/ERA5, we next presented a k-means clustering method

using seven years of the reanalyses (Aug-Oct 2014-2020) to examine the multi-year seasonal patterns and trends. While the humidity profiles are distinguished by both variations in total humidity and vertical structure, the CO profiles show a more uniform range, being primarily distinguished by maximum concentration in the free troposphere. Spatially and temporally, we see consistent spatial and temporal variations in smoke and humidity distribution (total concentration and vertical structure)





and their correlations to one another, reflecting changing conditions through the BB season. Generally, the high-humidity/high-CO profiles dominate in August and September in the northeast of the study region, and high-humidity/medium-CO dominates in October, with consistently lower concentrations in the southwest of the region. There, the atmosphere is largely unaffected by the smoke transported in the AEJ-S in August, although by October the entire region considered (6°W-12°E, 5°S-21°S) is

influenced by smoke and humidity to some degree. The distributions of the co-varied $q$/CO profiles range from being fairly strongly dominated by one type (30-40% for a given CO/$q$ combination) to more evenly distributed between many profile types ($\sim 5 - 10\%$ per type) in different places and times, highlighting the variation in episodic transport over this region.

    With this work, we have established a framework which allows us to more completely understand the spatial and temporal variations over the SEA, not just in atmospheric structure of water vapor and chemical species, but also ultimately of the

radiative effects of the elevated water vapor signal working in concert with the absorbing aerosol biomass burning plume. By establishing the consistently good agreement between the ERA5 and CAMS reanalyses and the aircraft observations, we can use the reanalyses to describe the frequency of conditions over the SEA. The six k-means profiles each for CO and $q$ allow for a comprehensive range of conditions to be described, while still limiting the total number of permutations to a manageable number. Previous studies have demonstrated the importance of water vapor and BB heating in this region, and in further work

will use this framework to comprehensively assess the impacts of the vertical and spatial structural variance of this region. The range in profile types, even as the general high-smoke/high-$q$ pattern remains, will allow us to quantify the radiative effects of atmospheric humidity and aerosol in this region, both together and as separate components. These results have potential impacts on our understanding of general circulation in the region, large-scale subsidence, precipitation distribution and mixing of BB into the cloud-topped boundary layer. Overall, this will allow for a more complete and nuanced assessment of the

aerosol-radiation-climate interactions in this region.

*Code and data availability.*  The flight data used in this paper are publicly available at http://dx.doi.org/10.5067/Suborbital/ORACLES/P3/ 2016_V1, http://dx.doi.org/10.5067/Suborbital/ORACLES/P3/2017_V1, and http://dx.doi.org/10.5067/Suborbital/ORACLES/P3/2018_V1 for the 2016, 2017, and 2018 data, respectively. The codes used in processing 4STAR data may be found at https://doi.org/10.5281/zenodo.1492912. ECMWF reanalyses are available at the Copernicus Climate Data Store (https://cds.climate.copernicus.eu/) or the Atmospheric Data Store

(https://ads.atmosphere.copernicus.eu/cdsapp#!/dataset/cams-global-reanalysis-eac4 for ERA5 and CAMS, respectively, subset and ordered according to the parameters described in the text. The MERRA-2 products used are available online at https://portal.nccs.nasa.gov/datashare/ iesa/campaigns/ORACLES/.

*Author contributions.*  KP designed research, performed ORACLES/reanalysis comparisons, and produced the figures. EMW processed ERA5/CAMS reanalysis to provide the k-means cluster results, methodology, analysis, and relevant supplementary figures. COMA data

were collected and processed by JP. 4STAR data were collected and processed by SL, KP, and MK. HiGEAR data were collected and processed by SGH and SF. KP wrote the paper, in discussion with EMW and PZ; all other coauthors had the opportunity to provide feedback on the methods, manuscript, and results.



*Competing interests.* The authors declare no competing interests.

*Acknowledgements.* KP, EMW, and MG's time was funded by NASA ACCDAM grant 20-ACCDAM20-0087. The ORACLES mission was funded by NASA Earth Venture Suborbital-2 grant NNH13ZDA001N-EVS2. Like most field work, the ORACLES project was very much a team effort; we thank the ORACLES deployment support teams, the ORACLES science team (for a more complete accounting, see the author list and Table C1 in Redemann et al., 2021), and the governments and people of Walvis Bay and Swakopmund, Namibia, and São Tomé, São Tomé e Príncipe for a successful and productive mission. We thank Ju-Mee Ryoo for many helpful conversations and comments, and Stephen Broccardo for comments on earlier drafts of this work.



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
