# Peer review of "Vertical structure of a springtime smoky and humid troposphere over the Southeast Atlantic from aircraft and reanalysis"

_EGUsphere, 2023_

## Author Comment (AC1)

*We thank both reviewers for their detailed comments. We hope we have satisfactorily addressed each one, and have made some additional minor editorial revisions for readability, flow, and typos. Our responses to the reviewers' comments are in italics below, and the tracked-changes version shows all edits made (referenced in relevant comments).*

**RC1: 'Comment on egusphere-2023-2412', Anonymous Referee #1**, 18 Dec 2023

Review of « Vertical structure of a springtime smoky and humid troposphere over the Southeast Atlantic from aircraft and reanalysis ».

I find that this is a good and well written that can be published after only minor corrections.

*We thank the reviewer for their comments and review.*

Page 5 line 23 : What does « incorporated » mean in this context? Is it assimilation ?

*We have clarified this passage with a better reference to Hersbach et al 2020, which discusses the datasets which are/not assimilated in ERA5, thank you for catching this.*

Page 8 : Was there some space-time interpolation needed to get the reanalysis on the same measurement locations as the ORACLES ? If so, what was it ?  The co-location part of the method is a bit unclear to me.

*Thank you for this comment and for catching this oversight. We used a nearest-neighbor collocation as in Pistone et al. (2021), and have added more detail on this to Section 2. The 3-dimensional nature of the reanalysis, plus the variability in profiles (some of which were ramps covering significant distance) led us to simplify the process in this manner.  We note that we found no dependence of the obs-reanalysis relationship (or goodness of agreement) on either spatial or temporal differences.  A version of Figs 2-4 focusing solely on the square spirals, i.e. more spatially constrained profiles, also showed much the same result, albeit with better overall spatial collocation across the profiles, due to the nature of the ramp maneuvers. We use both types of profiles in the main analysis as one-third of the ORACLES profiles were spirals versus ramps, and utilizing a larger portion of the dataset allows for more robust statistics.*

Page 8,  line 20 : reference needed for underestimation of CO by CAMS (e.g. Inness et al 2019) or other

*This sentence refers to the result shown in Figure 2 (as well as in Figs 3 and 4), with the CAMS vs observed comparison showing a slope <1. Following this comment and one from Reviewer 2, we have added additional supplementary figures to better illustrate the CAMS underestimation that we describe here, added the fits to Figs 2-4 for better illustration, and additional description from other references as in additional comments below.*

Fig2 caption:  change scatters to scatterplots

*We have made this edit.*

Table 1 : What accounts for the improvement in water vapor correlation from 2016-2018 in CAMS  but at the same time, a decrease in CO correlation ?  The q vs q correlation suggest that the dynamics and physics are working well in ERA5/CAMS but something has gone wrong with the CAMS CO. The correlation $R^2$ has fallen and is possibly not even significant. I think you need some significance test on these correlations. What is the explanation for this fall and for the low correlation (0.55). It there a

problem with the long-range transport in CAMS. Maybe a contingency table would be helpful to see if the plumes were captured but at the wrong location or the wrong time, or where nothing was seen at all with CAMS.

*While the focus of this paper isn't on the assimilation algorithms used in the reanalyses studied, and a full diagnosis would be beyond our current scope, there are a couple of possible explanations for the correlation variations, which we had intended to convey in our text (starting p8L32). First, there are substantial differences in deployment location resulting in different parts of the plume being sampled (i.e., farther from emissions sources means more opportunity for the reanalysis to diverge from the observations for a given emission inventory). The range in observed values may also produce lower $R^2$ values because, for example, the range in CO values is simply smaller in October. Finally, the BB conditions applied throughout the burning season may also be variably accurate (e.g., if the CAMS emissions inventories applied from the source are more in agreement with observations earlier in the season compared with October, which sees overall less emissions and the onset of moist convection, this will result in poorer correlation with the observations).*

*We did present the statistical significance of the values in Table 1 in the caption, where we indicate that the p-values were low (<0.001) for all correlations in the table. We agree that robust statistical significance is not always meaningful in physical terms, as suggested by the fact that the low MERRA-2 correlations also have low (i.e., significant) p-values, though obviously also have low $R^2$ as well. These points have been clarified in the body of the text (p. 9).*

*We would like to take this opportunity to note that in preparing this revision, we noticed an error in that the statistics presented in Table 1 (for 2017 and 2018) were for the full dataset rather than south of 5S as was stated in the text. The data in Figs 3 and 4 were (and are) as described. Reassuringly, and as was already stated, the numerical results are largely consistent for the two data (sub)sets; the primary difference in the updated table is a poorer $R^2$ in MERRA-2 for the 5S subset and, of course, fewer datapoints overall. We apologize for this error; it has been resolved in the revision.*

*A more comprehensive analysis of trajectories of individual airmasses would indeed be interesting, but we consider it beyond the scope of this particular work due to difficulty in distinguishing the sheer numbers of "individual" plumes in the region. Rather, as addressed in a later comment, we consider the region as containing a single plume of varying magnitude, and focus on the variability in magnitude and location of the persistent smoke plume over the region as it is made up of airmasses of mixed origins and ages. Figs 2-4 show that there is no common instance of an aircraft-observed plume not showing up at all in CAMS (i.e., there is no cluster of orange points below the 1:1 CO line, with the exception of a few in 2017 which are already fairly low-smoke conditions). That said, the "right airmass, wrong place" condition (specifically regarding vertical location) is a known issue with MERRA-2, as discussed in our text, with more detail provided in Pistone et al., 2021.*

Page 17, line 18 : Least-> lowest

*Done*

Page 20, line 2 : the most humid profiles (q1 and q2) are the most frequent

*This sentence has been reworded.*

Page 20, Line32 : it would be clearer to write :  the air at the altitude of the jet appears to originate from higher altitudes to the north of the jet more frequently that from the continent  directly to the east.

*This passage has been clarified.*

Page 22 line 20 : The background/smokeless case is probably not 100% smokeless and therefore you are capturing a real increase at altitude due to the sampling of long-range transport of plumes.  In the free troposphere there can be several layers with higher or lower mixing ratios of CO.

*We agree that the background case is probably not 100% smokeless. However, it would be puzzling if the CO actually peaked at 400hPa over the ocean, given the general extent of the continental boundary layer up to a typical maximum of 500hPa, which corresponds to the typical maximum of the AEJ-S. This combined with positive pressure velocity (downward motion) over the oceanic region suggests there is no direct pathway for continental smoky airmasses to reach these higher altitudes, and that 400hPa should be closer to the global background values for CO, which we expect to be lower than air masses influenced by the continental source region.  Regarding the multi-layered nature of the free troposphere, we acknowledge that this is a limitation of our approach: by selecting only 6 cases to be representative, and by using the coarser resolution of CAMS, we are not able to capture the true layered resolution of the BB plume. However, our goal with this study is to simplify the complex and varied atmospheric states in this region, to a manageable number of case studies. We have clarified these points in this passage of our text (track-changes p.24).*

Page 22, line 27 : with only a slight decrease in magnitude – does « magnitude » here refer to the magnitude of the discrepancy ?

*Exactly; that reference shows a consistent low bias from approximately 200hPa to 550hPa for the Windhoek site, with a small improvement in the agreement below that level. We've revised the text to clarify this point (p.24).*

Page 22 line 29, I thought profile 6 was showing an increase in CO with altitude.  Isn't it unlikely that it is a constant-CO atmospheric structure so there are always small increases throughout the free troposphere depending on the transport of different airmasses in different layers.

*This comment is similar to the one two comments above, so we hope we have clarified this point with the above response. Due to the general circulation patterns, the sources, and the previous discussion of known biases in CAMS, we believe that this profile is effectively a smokeless case, i.e. one which has not been significantly influenced by the continental BB plume but rather is fairly uniform "background values" (with the caveat of course that there are always variations in reality; these will likely be minimal given the CAMS resolution considered). This is again a limitation of our approach which attempts to condense the full range of observations into a limited number of case studies. We've added text in this paragraph to clarify these points, as well as a new supplemental figure showing the statistical range of the profiles classed into each k-means profile type (Fig S5; track-changes p. 24).*

Page 23, Line 24 : how would the frequency of the humidity profiles have any effect on the amount of smoke available for transport ?

*Thank you for this comment. We meant to convey that the frequency of humidity profiles and the smoke become somewhat decoupled from one another later in the season, not that the former influences the latter; this passage has been reworded to better clarify this point (track-changes p. 25).*

Page 23, Line 25 : The biomass burning plume, (which one) ? Or biomass burning plumes (in general) ?

*We use the term "plume" throughout to mean the generally persistent seasonal signal of biomass burning aerosol and associated combustion products (such as CO), following the nomenclature used in other works (e.g. Eck et al 2013; Redemann et al, 2021, and the references therein), although, as we intended to convey here, the "plume" certainly varies episodically (spatially and temporally). We have revised this passage for clarity (track-changes p. 25).*

Page 24, Line 25 : Please rewrite this sentence: This does not mean that this more remote region is not influenced by the biomass burning plume (indeed, this sector contains St Helena Island, whose smoke and humidity vertical structure are studied by Adebiyi et al. (2015) and others, and the Ascension Island observatory is located to the west of the northwesternmost sector we consider, i.e. even more removed from the source); but the smoke there is more episodic.

*I believe this comment refers to the sentence in the previous comment; we have revised this sentence per the above.*

Page 28 : I do not see the link between the first paragraph of this discussion and the results presented in the article. You need to integrate or link the discussion with the results section to make this part relevant to what we have just been reading about, otherwise it seems like an afterthought.

*Thank you for this comment. This section was intended to place our work in the motivational context, i.e., the "why" rather than just the descriptive "what," the latter of which makes up the bulk of the present manuscript, but evidently we needed to make those connections clearer. We have revised this section (track-changes p. 30) and added some text to the first paragraph of the introduction (p. 2) to better communicate this framing.*

**RC2: 'Comment on egusphere-2023-2412', Anonymous Referee #2**, 08 Jan 2024

Review on preprint "Vertical structure of a springtime smoky and humid troposphere over the Southeast Atlantic from aircraft and reanalysis" by Pistone et al.

This manuscript proposes a climatological characterization of water vapor and biomass burning plumes in the south-east Atlantic area, with the overarching aim of gaining a better understanding of the radiative impact of aerosols. Therefore, the authors rely on model reanalyses, constrained by data assimilation (ERA5 and CAMS). As observations and simulations of aerosol concentrations are difficult to exploit due to higher uncertainties, the diversity of the compounds making up aerosols and the complexity of the physical parameters influencing AOD (species involved, optical properties but also aerosol size distribution, mixing state…), the authors use CO concentrations as a proxy for the signature of fire plumes.

To support this choice, the authors first demonstrate the ability of the models used to reproduce water vapor, as well as the general structures of CO and AOD, and then demonstrate the good correlation

between AOD and the total CO column in observations and simulations. Using k-means clustering of the CO and q profiles mapped onto their 2 principal components, the authors then show that the variability is well represented by 6 clusters of q and 6 clusters of CO. Finally, the number of profiles in each cluster for each month of the studied time period is analyzed.

This manuscript reports on a milestone in a wider study, since it does not go as far as analyzing the radiative impact of aerosols for the identified typical situations. Nevertheless, the work is original, rigorous and well presented. Most of the figures are very clear and the text well written.

I recommend publication, and suggest only minor revisions, listed below.

*We thank the reviewer for their review, and hope we have sufficiently addressed the specific comments below.*

 Minor comments:

Abstract.: it would be helpful to have a summary of the main features identified in the clusters.

*Thank you for this comment. It was a struggle to get our abstract down to the 250-word limit now required by ACPD, and much desired detail had to be excluded. If it is okay with the editorial team, we have added a more descriptive summary in the last paragraph of our abstract to address this comment.*

Section 2:

The different sections should include some discussion on the uncertainties for each dataset (with numbers).

*Thank you for catching this oversight. While Section 2.1 already included some description of the measurement uncertainties, we have added more text and references in the relevant paragraphs. In all cases, the observational uncertainties are much smaller in magnitude than the ranges seen in the observation-reanalysis comparisons (which of course have no uncertainties associated with them, other than uncertainties stemming from spatial and temporal resolution).*

For the reanalyses: I think CAMS also assimilates IASI CO observations. I would also be important to detail the biomass burning emission inventory used in the simulations, as well as the potential corrections. I think that the GFAS inventory is used. In the reference paper, Kaiser et al. (2012) recommend applying a factor of 3.4 on BB emissions for aerosols. Is it the case in the reanalyses used here? This could partly explain why the underestimate in AOD is lower than the underestimate in CO mentioned in Section 3.1.

Kaiser, J. W., Heil, A., Andreae, M. O., Benedetti, A., Chubarova, N., Jones, L., Morcrette, J.-J., Razinger, M., Schultz, M. G., Suttie, M., and van der Werf, G. R.: Biomass burning emissions estimated with a global fire assimilation system based on observed fire radiative power, Biogeosciences, 9, 527–554, https://doi.org/10.5194/bg-9-527-2012, 2012.

*Per the Inness et al (2019) reference, the current CAMS reanalysis assimilates only MOPITT and not IASI CO (they explain that it was included in a previous reanalysis product, but produced discontinuity). Per the same reference, CAMS does indeed use GFAS v1.2 for the biomass burning emissions inventory, which*

was updated from the v1.0 described in the Kaiser et al., 2012 reference. The scale factor for BC is regionally determined and applied (Inness et al, 2019, Section 2.1.1).  There are a number of factors which likely contribute to the quantitative discrepancies between the observations and reanalysis, for aerosols this is particularly acute since necessarily the satellite assimilations are of total and total-column aerosol optical depth, and the model starts from twelve speciated aerosol bins, which are then aged, processed, and integrated and summed (assuming certain optical properties) to get something comparable to satellite-observed AODs. Several additional papers could likely be written on the relative contributions to these discrepancies, but to keep the scope of the current work manageable, we hope to limit the discussion specifically to the "what" rather than the "why" of these differences.

Section 3:

For model evaluations, it would be helpful to provide the statistics of the comparisons in terms of mean relative bias for q and CO in Table 1 (as for AOD in Table 2).

*We have added an additional supplementary figure (new Figure S3) to better illustrate the AOD, CO, and q differences for each deployment.*

From Figures 2-4, it is not clear to me that CAMS only underestimates large CO values (authors mention values > 300ppbv). Showing the mean simulated and observed profiles, and the corresponding mean bias, would be interesting. Also, models tend to simulate wider plumes due to too large dispersion. Could this explain an underestimate of maximum values (in addition to the dilution due to the resolution), and an overestimate of background values in some areas? A general discussion of the CAMS model performance in simulating the LRT of BB plumes in the literature would be helpful (could also be in section 2).

*In Figs 2-4, the more dramatic underestimation at higher CO values is seen in the difference in slope of the plume level data compared to the 1:1 line. We have added the least-squares fits (from Table 1) to the data in these figures to better highlight this feature, and an additional composite supplemental figure (new Figure S1) showing a subset of aircraft profiles and the corresponding reanalysis profiles, which illustrates the underestimation at the altitude of maximum plume.  There does not appear to be a particularly overly-dispersed plume in CAMS (i.e., we see no corresponding CO overestimation at other altitudes outside the plume maximum; Fig S1), but we acknowledge that ~1km vertical resolution considered may not be optimal for diagnosing that particular feature.  Regarding the CAMS model performance question, it is difficult to answer beyond the scope of this paper (i.e., "performance" is usually identified by comparison to observations, which are limited in many remote regions), but by presenting this work, we hope we have addressed that question for the relevant SEA region and time. We have revised the discussion in the (new) Section 3.1 (p.6) and Section 4 (p. 24) to better convey these points.*

Section 3.1:

Would it be possible to calculate the relative aerosol mass below the aircraft to evaluate the possible overestimate in AOD if the full column is considered?

*Because of the instrument design and aerosol profiles, there is no way to actually quantify below-aircraft aerosol; we would only be able to assume some constant or weighted PBL AOD. In the ORACLES dataset, there is an above-cloud AOD flag (determined according to inlet-based gradients where available), which*

is available over more spatial range than the profiles shown here, but does not allow for characterization of the vertical structure that the aircraft profiles show. More details on the full-column AODs available may be found in LeBlanc et al (2020), doi: 10.5194/acp-20-1565-2020. We also note that for the aircraft profiles considered in the current work, the observed/reanalysis comparison (Fig 6, colorbar) shows no dependence on altitude (i.e. is not indicative of a systemic below-aircraft AOD being missed).

Among the uncertainties in the simulated AOD, the authors could also mention the size distribution and the aerosol mixing state…

*This has been made more explicit in Section 2.2 (track-changes p.6).*

Regarding the impact of hygroscopicity, what would be the associated variation in AOD? As this been evaluated in CAMS?

*That is an outstanding question; to the best of our understanding, aerosol hygroscopicity evolution is not included in the CAMS aerosol scheme. Previous studies using the ORACLES observations attempted to qualitatively assess these impacts, and despite the well documented water content presented here, there is relatively low \*relative\* humidity (with approximately 70% of observations at RHs below 60%, and 98% of observations above the PBL having RH<80%; Pistone et al 2021) seen in the aircraft data. Thus, the humidification effects are likely not dramatically impacting aerosol optical properties. Shinozuka et al., 2020 estimated that the effect of aerosol hygroscopic swelling on extinction was fairly minimal in the free troposphere, with an ambient RH/dry ratio of less than 1.2 for 90 % of measurements, suggesting the same may be true for AOD. Given the complexity of the various factors influencing aerosol variation (and discrepancies of reanalysis compared to observations), this is beyond the scope of the present study, although we are certainly aware of and interested in future work on the magnitude and potential impacts of these processes.*

Section 4:

The figures in the supplementary material really help to understand the method, but the legends need to be revised as they are far too vague and not self-explanatory (especially figures S3-S8). It seems to me that the principal components themselves are not shown (cf. l.12, p.17).

*We have revised the PC supplementary figures and added more detail in the figures and their captions to clarify the method.*

I am surprised that the shape of CO profile is very similar for all clusters, unlike that of water vapor, and I was wondering if similar profile shapes were observed during the ORACLES campaign. Could this shape be due to the data assimilation using satellite instruments that lack vertical sensitivity?

*This is indeed an interesting point. The key difference between CO and q is their source: while CO in this region is an indicator of air masses originating on the smoky continent, air masses of high q in this region can either be of smoky (and humid) continental origin, or from the ocean surface/boundary layer. Thus while it may seem non-intuitive that these closely correlated parameters show different vertical shapes, it's essentially showing the same thing: a variation in plume altitude and vertical extent, with the distinction that the profiles always have a more-humid and less-smoky boundary layer below. This has been clarified in the text (track-changes p.23).-24 We also acknowledge that the more coarse vertical resolution of CAMS likely limits how much the shapes can vary.*

*Similar profile shapes were observed in ORACLES, a selection of which was shown in Pistone et al 2021 Fig 8. Per this and some other reviewer comments, we have added additional supplementary figures showing the aircraft CO profiles (Fig S1) as well as the ranges of reanalysis CO and q profiles corresponding to each k-means cluster (Fig S5).*

Do the CO profiles corresponding to the cluster q4 also exhibit a signature from a transport of air masses from another area (linked to the dry intrusion above the PBL)?

*That's an interesting question; while our previous trajectory analysis established the humid air mass origin from the continent (Pistone et al 2021, Figs 16-17), we have not explicitly performed any case studies on the origin of the drier "gap air". However, Figure 9 in the current work suggests large-scale patterns of southwesterly winds at 600-700hPa, as was described in the text (p.20L18 in the original submission). This has been clarified in the following paragraphs (track-changes p. 21,23; this section has been slightly restructured for better flow). The CO profiles corresponding to q4 are largely C3 and C5, which are profiles which show a higher-altitude CO peak compared with C4, which is consistent with this picture.*

p.22, the authors discuss the cluster C6 and a possible overestimate in background values: cf previous comment for Section 2.

*I think the Section 2 comment to which the reviewer refers is the one on the assimilation/BB inventory used in CAMS. I hope we have adequately addressed this in the above response to that comment.*

p.23: l. 1-2: the authors state that the profiles from the k-means is consistent with the observations. I may have missed the information but I think it would be useful to include more precise comparisons in terms of profile shape, for CO in particular since uncertainties in the simulations are larger. This could be essential for later use in a radiative transfer model assessing the radiative impact of aerosols.

*We have added supplemental figures showing the distributions in profiles for each k-means cluster for both q and CO (Fig S5), and agree that the ranges shown here will be quite instructive when establishing the RT calculations.*

The authors mention that clusters corresponding to lower CO concentrations include both background profiles and more episodic transport (p.23). Would the number of clusters help differentiate profiles with no BB signature and those with a smaller plume?

*We experimented with a range of clusters (between 4 and 9) to try to determine if there was an optimal number before settling on 6.  For a cluster analysis of >6 clusters, there is no additional information added in terms of new shapes or distributions; instead the extra clusters added are intermediates in magnitude between the existing clusters.  Thus, we chose 6 as a balance between capturing the full range of structure, and not having too many subdivisions so as to confound the results. As in the comment above, the ranges of values in each cluster will allow us to evaluate the magnitude and significance of these differences.*

The discussion of the temporal variability of the number of profiles in each cluster could be linked more explicitly to the variability in fire activity.

*We have added more discussion of the seasonal changes in fire patterns (track-changes p.25).*

Discussion:

p.28, l. 23-24: The authors state that the case studies of Cochrane et al. 2022 are included in the climatology presented in the paper. To what clusters do they correspond?

*The particular case studies shown in the Cochrane work are included in the aggregate comparisons of Figs 2 and 3. The 9 specific profiles they present correspond to C2, C3, or C4, and q2, q3, or q4 following the k-means method.*

Technical corrections:

p.5, l. 6: scattering aerosol extinction (no need for a capital A)

*Thanks, fixed.*

Section 3: only a subsection 3.1 for aerosols. Maybe include another subsection for CO comparisons?

*This may be stylistic as much as anything, but point taken that another subsection makes sense here.*

p.8, l.24: missing verb?

*A small one, but yes, thank you, fixed.*

Supplemental figures need more precise legends.

*We have updated the relevant supplemental figures with more detailed captions and axes.*

[revised manuscript text omitted]

5 in physically thicker clouds with higher liquid water path (LWP) (e.g., Wilcox, 2010; Wilcox et al., 2016) and thus higher cloud albedo. In a modeling-based study, Ackerman et al. (2004) modeled the influence of above-cloud water vapor using several case studies informed by field measurements, and concluded that the cloud liquid water path response to aerosol (i.e., increasing LWP due to precipitation suppression) had a much stronger response in the presence of overlying water vapor than under dry conditions, due to changes in cloud-top entrainment. Deaconu et al. (2019) used satellites and reanalysis to conclude

10 that absorbing aerosols in the SEA increased cloud optical thickness and liquid water path, and lowered cloud top altitudes. In a recent study using large eddy simulations, Baró Pérez et al. (2023) found that the moisture in the SEA BB affected the evolution of the underlying cloud field, in addition to radiative effects of the humid BB plume.

[revised manuscript text omitted]

For sections where the observations are compared with reanalyses, the 1Hz aircraft observations are averaged within 100m ($\pm$50m) of the reanalysis pressure levels. The comparisons with the ECMWF reanalyses are first paired to the closest reanalysis timestep to the profile mean time, then the nearest-neighbor latitude and longitude to the mean profile locations so as to maintain consistency with the later sections focused solely on reanalysis fields. We note that the results presented here are consistent for subsets of the data considering either solely square spiral profiles (i.e., the more spatially localized profiles) or profiles with the greatest temporal collocation (<1h).

[Figure]

**Figure 1.** Regional distributions of observations and the spatial subdivisions used. (a) From Pistone et al. (2021), the locations of ORACLES-2016 aircraft profiles (orange circles), the regions used for the aircraft-based analysis (blue) and the reanalysis study (lavender). In (b) ORACLES-2017 and (c) ORACLES-2018, more northern profiles were sampled. For ease of comparison, these data (green grid) are divided at $5°$S to aid comparison with 2016 (blue grid). Note also the larger westward range of observations in 2017. (d) Summary of the different subdivided regions considered in this and other works. The stars show the sites of previous island-based observations at St Helena Island (yellow star) and Ascension Island (blue star). The thicker black lines are the SEA divisions discussed in Section 4, on a nominally $6° \times 6°$ grid which is slightly uneven north-to-south to align with the CAMS spatial resolution and to isolate the AEJ-S latitudes.

**3  Agreement between reanalyses and aircraft observations**

In assessing the representativeness of the reanalyses and their agreement with the ORACLES observations, we  focus on the parameters of specific humidity, the biomass burning tracer CO, and both column and inlet-based aerosol extinction.

**3.1  Water vapor and BB tracer (CO)**

We begin our analysis with specific humidity $q$, a meteorological field, and CO, a biomass burning tracer. We choose CO as our
5 BB tracer as the modeling of chemical species is subjected to fewer uncertainties than is aerosol modeling. Aerosol as modeled in the CAMS reanalysis considers aerosol speciation, processing, lifetime, and removal for 12 separate aerosol components, while the assimilated observation is column total AOD from satellites (Inness et al., 2019). Because of this, we expect CO to have a more straightforward and accurate representation than the fields of aerosols themselves, for our purposes of assessing and characterizing atmospheric distributions over time. We note that the lifetime of CO in the atmosphere is 1-4 months (Szopa
10 et al., 2021) and thus may  result in accumulation to a higher background value over the span of a given biomass burning season (3 months), but as we show in Section 3.2, the observed aerosol-CO relationship does not vary across the three airborne campaigns.

Previous work in Pistone et al. (2021) showed that the ERA5 reanalysis captured the vertical structure and location of the humid plume remarkably well in September 2016 (there reported as an observed/ERA5 correlation of $R^2 = 0.79$ for observations
15 above 2km over all flights). ERA5 performed better than MERRA-2 in terms of this direct comparison (observed/MERRA-2 $R^2 = 0.40$), due to a known vertical velocity (excessive subsidence) issue in the latter (e.g., Das et al., 2017). Figure 2 shows comparisons between (left, middle, right) ERA5, CAMS, and MERRA-2 versus the observed values from the 2016 ORACLES deployment for (top, middle, bottom) specific humidity, CO, and the relationship between the two. The CAMS reanalysis shows a similar pattern in specific humidity and agreement with the observations ($R^2 = 0.84$ for $z \geq 2$km), despite its lower
20  resolution (Section 2) giving roughly half as many matches compared with ERA5. In all the reanalyses considered, the largest reanalysis-observation discrepancies occur near the top of the boundary layer ($\sim$1-2km).

For the CO comparisons (Figure 2, middle row), there is similar good agreement overall between CAMS and observations ($R^2 = 0.75$ for $z \geq 2$km) although the reanalysis tends to underestimate these values for higher observed CO ($\gtrsim$ 300ppb; this is also seen in individual profiles, as shown in Supplementary Figure S1). The overall slope between the CAMS and observed
25 CO is 0.79  (i.e., showing underestimation), compared with 0.98 for the CAMS $q$ versus observed $q$. Still more variability is seen in the direct comparison with MERRA-2 (CO observed/MERRA-2 correlation: $R^2 = 0.21$). This general pattern is consistent over all three ORACLES deployments (Figures 2, 3, and 4), and is also consistent with Pistone et al. (2021), likely due to the aforementioned issue with vertical plume displacement. In other words, MERRA-2 tends to underestimate the higher altitude points while sometimes overestimating lower-altitude points for both CO and $q$ relative to aircraft observations, while
30 still preserving the relationship between CO and $q$ for MERRA-2 overall (e.g., Figure 2, bottom right).

The data from August 2017 (Figure 3) and October 2018 (Figure 4) are largely consistent with the picture from September 2016, with a few notable differences. First, the correlations of observed versus CAMS or ERA5 water vapor are slightly better

than in 2016 and substantially better for MERRA-2 (Table 1, with the caveat that there are fewer profile matches for these years). However, the CO observed versus CAMS is a somewhat poorer match both in terms of the $R^2$ values and the slope for the latter deployments.  (All correlation coefficients are statistically significant at the $p < 0.0001$ level.) This difference in correlations between the three deployments may be partly explained by seasonal evolution of the smoke plume; specifically, higher CO observed in August

5   2017 results in a slope which is skewed low due to the CAMS high-CO underestimation, and the lower CO observed in October 2018 results in a less robust (i.e., lower) $R^2$ value due to a smaller dynamic range in CO values. It is also possible that there are differences in the emissions inventory assimilated into CAMS for each month of the BB season, although such diagnostics would be beyond the scope of this paper.

[revised manuscript text omitted]

The conditions captured in the 6 k-means-defined CO profiles (Figure 8) are somewhat less varied in structure than those for $q$. While it may seem counterintuitive that two parameters so closely correlated exhibit different vertical structures, this is due to the (above background level) CO in this region indicating an air mass origin over the smoky continent, whereas $q$ in this region may originate either on the smoky (and humid) continent or from evaporation from the ocean surface. Essentially, despite their different shapes, the two sets of k-means profiles show the same feature: a variation in (humid smoke) plume altitude and vertical extent, always with a more-humid and less-smoky boundary layer underneath. Regardless, for the CO profiles there is still variability in BL CO ( ranging between 63 to 178ppb for 950-1000hPa, with mean values nearly constant for a given profile), but the primary difference is the range at in the free troposphere, from high concentration (~ 400ppb) to a background value of 60-70ppb. The peak concentration is seen either at 700 hPa (C3, C5) or 800 hPa (C1, C4), with C2 nearly uniform across this range (~ 2 − 3.2km). This is lower in altitude than the typical maximum of the south African Easterly Jet (AEJ-S)  at 600-700hPa (Adebiyi and Zuidema, 2016; Pistone et al., 2021; Ryoo et al., 2022); the coarse pressure level resolution of the CAMS reanalysis  is likely a limiting factor, plus there is a degree of vertical

subsidence with time over the region ($\sim 50-80$hPa/day) which is seen in both the reanalysis and the observations, as was previously shown.

We note that a curious outcome of applying this method to CO profiles is that profile C6 (the background/smokeless case) shows slightly increasing CO with increasing altitude, from 65ppb at 1000hPa to 88ppb at 400hPa. We suspect this may not be a real characteristic of the atmospheric structure  (as the continental outflow peaks around 500hPa and there is large-scale subsidence over the ocean, it would be puzzling for there to be a maximum in CO above this altitude), but rather may be an artifact resulting from limitations in the CAMS satellite assimilation. Previous studies have documented low biases in CO in the lower and middle troposphere when compared with aircraft profiles (e.g. Inness et al., 2019, 2022). Specifically, Inness et al. (2019) showed consistent low-biases of 10-20% in CAMS reanalysis CO between 600hPa and the surface, for all airport sites considered. While the Windhoek, Namibia (i.e., closest geographic to our study area) site showed  a fairly constant low bias through these altitudes  (with a slight improvement in the reanalysis-observation bias near the surface), other sites show a pronounced increase in the negative bias at the lower altitudes relative to higher in the troposphere. The complicated atmospheric structure in this region makes it plausible that this may also be occurring in the present case. In other words, profile C6 is not showing a true decrease in CO with $z$, but rather an unphysical artifact in an actually-constant-CO atmospheric structure. Manual inspection of some C6 profiles indicate the presence of a small CO layer in some cases, and a more uniform profile in others,  but the majority of profiles classed as C6 fall close to this profile structure (Figure S5). Taken together, the evidence suggests this result may be a limitation of our k-means classification method; by aiming to identify a limited number of profile conditions, some less-smoky profiles  with somewhat varied structure end up collapsed into one  effectively smokeless case (i.e., close to background CO values at all altitudes). Inness et al. (2022) also mention that the underestimation of CO is a common problem in many atmospheric chemistry models, although we note that the minimum CO observed (i.e., background) by aircraft during ORACLES is around 60ppb, closer to the low-altitude values in this profile, which suggests instead a potential overestimation in CAMS CO at higher altitudes rather than an underestimation at the lower altitudes. (A full quantification of the sources of uncertainties in the reanalysis is beyond the scope of the present work.) Nonetheless, the  demonstrated presence of an effectively smoke-free atmospheric profile (C6) is consistent with expectations in this region, and is a useful case study towards our goal of a broader, and necessarily simplified, classification of the complex and varied atmospheric states in this region. We will leave the specific implications of the particular shape and incidence of C6 for a future work. Overall, this k-means method produces a range of profile types which are consistent with the observations across the SEA.

The CO profile classifications (Figure 8) are in many ways more straightforward than those of $q$, but their spatial distribution is more complex. Compared with the $q$ distributions in Figure 7, the CO distributions are somewhat less continuous throughout the season, although some general patterns are still clear. The northeasternmost sector (4.5-10.5°S, 6-12°E) has the highest percentage of high-CO profiles, although the incidence of these profiles (C1 and C2, respectively) shows a seasonal trend opposite to that of $q$. Specifically, CO concentration is greatest in August and decreases in October, contrasting the increase in $q$ over this time. The profile frequencies for that sector are 41% (C1 only) and 85% (C1 or C2) in August; in September, C1 alone drops to 12% frequency, but the summed frequency of C1 plus C2 is still 73% overall. By October, less than 0.5%

35  of profiles are classed as C1, and only around 19% are classed as C2. In other words, even as the water vapor increases from August to October, the CO (and by proxy, smoke) decreases dramatically, providing an opportunity to better examine the impacts of the dynamic range of the two profiles together.

There are other features which distinguish the distribution of CO profiles from that of the  $q$ profiles. There is a notable lack of a consistent trend across a three-month season within a specific region, particularly south of 10°S, in contrast to the

5  consistent increase in $q$ (and increase in associated $q$ profile types) in all regions. Instead, the CO profiles often peak (or trough) in September, with lower (or higher)  incidence in August and October (e.g., C3 in the northern and middle latitudes far from coast, and C2 near coast); alternatively, in other sectors the CO values increase (e.g., C5, south of 10.5°S) or decrease (e.g., C6, in the south) through the season. CO profiles in the middle and southern sectors are more likely to be classed as one of the more smoky profiles in September than in August or October, despite the progression of more- to less-

10  smoky in the more northern regions. By September, almost all profiles north of 15°S have some CO between 500-900hPa, while south of 15°S, 12-35% of profiles are still classified as "unpolluted" (C6), again likely due to intrusion of southwesterly airmasses (Ryoo et al., 2021). Some expected seasonal patterns are still evident; despite significant frequency of smoky profiles especially in September, by October, the middle-west region is dominated by the cleaner profiles (72% here are C6 or C5, i.e. one of the two least smoky profiles), and the northwestern and middle regions are almost half (45% and 48% respectively) C5

15  or C6. This reflects the decreased amount of smoke available for circulation later in the season  due to a combination of seasonal factors, e.g., seasonal changes in burning patterns, namely the southeastern progression of detected fires (e.g., Redemann et al., 2021); a potential an increase in smoldering versus flaming burn conditions (e.g., Eck et al., 2013) into October-November; and the more frequent precipitation events at AEJ-S latitudes in October (Ryoo et al., 2021). These changes in BB conditions occur as the high-humidity profiles become more frequent in these same regions  over this time

20  (Figure 7), following the increasing incidence of moist convection over the southern African continent (Ryoo et al., 2022). A strong AEJ-S condition also persists in all months (Figure 9). , ensuring these continental airmasses move over the SEA throughout this season. Taken together, this does not mean that  the more remote areas of the SEA region are never influenced by the seasonal biomass burning plume (indeed, this sector contains St Helena Island, whose smoke and humidity vertical structure are studied by Adebiyi et al. (2015) and others,

[revised manuscript text omitted]

**Figure S3.** Cumulative distribution functions showing the agreement between collocated reanalysis-observation comparison points (differences, left, and in absolute values, right), for each deployment year. The right side shows the good absolute agreement between reanalysis and observations, especially for the vertically resolved values (i.e., CO and water vapor), and the left side shows the tendencies of these distributions towards over- vs underestimations (e.g. slight overestimation in AOD in CAMS, and slight underestimation in CO, with the exception of 2018.)

[Figure]

**Figure S4.** Gridded CAMS column total column carbon monoxide versus column biomass burning AOD (organics + BC) for 2017. August (blue) has a less steep slope than October (yellow) with September (orange) intermediate, showing the seasonal evolution of column values.

[Figure]

**Figure S5.** K-means classifications for $q$ and CO, showing the range (standard deviation) of the ERA5 and CAMS (respectively) profiles classified as that profile type.

[Figure]

**Figure S6.** Principal component analysis (left panel) showing the relative abundance of water vapor profiles in ERA5 in the phase space of the first and second principal components (PC1 and PC2, respectively) of all ERA5 water vapor profiles over the region. 'X' symbols indicate the centroid locations in the phase space of the principal components of a k-means clustering  analysis on abundance of water vapor profiles in the  phase space. Right panel shows the resulting water vapor profiles associated with each of the 6 clusters produced from the k-means clustering analysis, which serve as the canonical set of water vapor profiles over the region shown in Figure 7.

[Figure]

**Figure S7.**  Same as Figure S6 but for profiles of CO from the  CAMS simulations. Right panel shows the resulting CO profiles that serve as the canonical set of CO profiles shown in Figure 8

[Figure]

**Figure S8.**  Left panel shading is the same as left panel in Figure S6. 'X' symbols show samples selected from across the range of values of the first principal component for a constant value of the second principal component. Right panel shows the resulting water vapor profile corresponding to each 'X' symbol and illustrates the variability of the first principal component of the water vapor profiles , which is largely showing the variation in  the magnitude of the column water vapor concentration.

[Figure]

**Figure S9.**  As in Figure S8 but the right panel illustrates the variability of the second principal component of the water vapor profiles (right panel),  which is capturing the variations in upper-level q between the 500 and 800 hPa pressure levels relative to the lower-level $q$ below the 850 hPa pressure level.

[Figure]

**Figure S10.**  As in Figure S8 but the right panel illustrates the variability of  the first principal component of the CO profiles, which is largely showing the variation in  the magnitude of the column CO concentration.

[Figure]

**Figure S11.**  As in Figure S10 but the right panel illustrates the variability of  the second principal component of the CO profiles,  which is capturing the variations in upper-level CO concentration between the 500 and 800 hPa pressure levels relative to the lower-level CO concentration below the 800 hPa pressure level.